# Mycobacterial resistance to zinc poisoning requires assembly of P-ATPase-containing membrane metal efflux platforms

Yves-Marie Boudehen[1,4,7], Marion Faucher[1,7], Xavier Maréchal[2,5], Roger Miras[2], Jérôme Rech[3], Yoann Rombouts [1], Olivier Sénèque[2], Maximilian Wallat[1,6], Pascal Demange[1], Jean-Yves Bouet[3], Olivier Saurel[1], Patrice Catty[2,8], Claude Gutierrez [1,8] & Olivier Neyrolles [1,8] ✉

The human pathogen *Mycobacterium tuberculosis* requires a $P_{1B}$-ATPase metal exporter, CtpC (Rv3270), for resistance to zinc poisoning. Here, we show that zinc resistance also depends on a chaperone-like protein, PacL1 (Rv3269). PacL1 contains a transmembrane domain, a cytoplasmic region with glutamine/alanine repeats and a C-terminal metal-binding motif (MBM). PacL1 binds $Zn^{2+}$, but the MBM is required only at high zinc concentrations. PacL1 co-localizes with CtpC in dynamic foci in the mycobacterial plasma membrane, and the two proteins form high molecular weight complexes. Foci formation does not require flotillin nor the PacL1 MBM. However, deletion of the PacL1 Glu/Ala repeats leads to loss of CtpC and sensitivity to zinc. Genes *pacL1* and *ctpC* appear to be in the same operon, and homologous gene pairs are found in the genomes of other bacteria. Furthermore, PacL1 colocalizes and functions redundantly with other PacL orthologs in *M. tuberculosis*. Overall, our results indicate that PacL proteins may act as scaffolds that assemble P-ATPase-containing metal efflux platforms mediating bacterial resistance to metal poisoning.

Transition metals are essential elements for all organisms. As such, they are the subject of fierce competition between microbial pathogens and their hosts. Infected hosts employ multiple strategies to restrict the access of invading pathogens to metals, a process known as "nutritional immunity"[1–5]. To counteract immune-mediated nutrient withholding, pathogens have evolved a variety of mechanisms to extract and incorporate nutrients, including metals, from their host's cells and tissues, a process recently referred to as "nutritional virulence"[6–11].

Although essential in trace amounts, transition metals are toxic when in excess, and their intracellular concentration must be tightly regulated. We and others discovered that immune cells exploit transition metals, namely zinc and copper, to intoxicate bacterial pathogens, and that metal efflux systems are involved in bacterial virulence[12,13]. Specifically, we showed that, in human macrophages, zinc accumulates in phagocytosis vacuoles containing the tuberculosis (TB) bacillus, *Mycobacterium tuberculosis*, and that the metal efflux pump CtpC, a member of the P-ATPase superfamily[14–16], is required for

[1]Institut de Pharmacologie et de Biologie Structurale, IPBS, Université de Toulouse, CNRS, UPS, Toulouse, France. [2]University Grenoble Alpes, CNRS, CEA, IRIG, Laboratoire de Chimie et Biologie des Métaux, 38054 Grenoble, France. [3]Laboratoire de Microbiologie et de Génétique Moléculaires, LMGM, Centre de Biologie Intégrative de Toulouse, CBI Toulouse, CNRS, Université de Toulouse, UPS, Toulouse, France. [4]Present address: Centre National de la Recherche Scientifique UMR 9004, Institut de Recherche en Infectiologie de Montpellier (IRIM), Université de Montpellier, Montpellier, France. [5]Present address: Evotec SAS, Campus Curie, Toulouse, France. [6]Present address: London School of Hygiene and Tropical Medicine, London, UK. [7]These authors contributed equally: Yves-Marie Boudehen, Marion Faucher. [8]These authors jointly supervised this work: Patrice Catty, Claude Gutierrez, Olivier Neyrolles. ✉e-mail: Olivier.Neyrolles@ipbs.fr

*M. tuberculosis* to resist zinc intoxication and multiply inside these cells[12]. Since then, a number of studies have reported convergent findings, pointing to a key role for metal scavengers or exporters in the virulence of fungal and bacterial pathogens[17,18], including *M. tuberculosis*[19–21]. Thus, metal detoxification machineries are major points of vulnerability in pathogenic microbes, and are promising targets for the development of novel antimicrobials[22–25].

Metal ions are hydrophilic and cannot cross biological membranes by passive diffusion. $P_{1B}$-ATPases are involved in the efflux of transition metals[15]. All P-type ATPases are integral membrane proteins containing six or more transmembrane (TM) helices[26,27]. The *M. tuberculosis* genome contains seven $P_{1B}$-ATPase-encoding genes, namely *ctpA-ctpD*, *ctpG*, *ctpJ* and *ctpV*, the characterization of which is still limited. In particular, the substrates of a few mycobacterial $P_{1B}$-ATPases are known from experimental evidence[20,28–32], and those of the others can be predicted from in silico analysis and the presence of conserved motifs[14–16,33,34]. CtpC was proposed by Padilla-Benavides et al. to be an exporter of manganese, albeit with an unusually low efficiency[35]. We proposed that CtpC exports zinc, based on the specific induction of *ctpC* by zinc, the selective accumulation of zinc in an *M. tuberculosis ctpC*-null mutant, and the high sensitivity of this mutant to zinc excess[12].

We had found that CtpC is encoded together with a 93-residue protein, Rv3269, in a zinc-inducible operon[12]. Rv3269 is predicted to contain a transmembrane domain and a metal binding motif. Intriguingly, several Cu⁺-ATPases are co-transcribed with copper metallochaperones that can bind and donate copper to their cognate P-ATPase transporter[36–38]. Copper metallochaperones are generally soluble proteins, although a membrane-anchored member of this family was reported in *Streptococcus pneumoniae*[39]. Based on these observations we hypothesized that Rv3269 might act as a metallochaperone for CtpC[23,24,33].

Here, we show that Rv3269, hereafter renamed PacL1 for "P-ATPase-associated chaperone-like protein 1", is a zinc-binding chaperone-like protein that plays an unexpected dual role in resistance to zinc poisoning. Its metal binding domain is required for survival only at high concentrations of zinc, consistent with a metallochaperone or chelator function, or both. However, PacL1 is required for survival even at low zinc concentrations, independent of its metal binding domain. We provide evidence that PacL1 additionally acts as a scaffold to stabilize CtpC at the plasma membrane in previously undescribed, P-ATPase-containing functional membrane foci that resemble functional membrane microdomains[40,41]. Our data show that CtpC and PacL1 physically interact at the plasma membrane and form high-molecular weight complexes. We propose to call these complexes "metal efflux platforms", highlighting the fact that several proteins of different substrate specificity assemble in a high molecular weight complex to fulfill a unique function, i.e., metal efflux in this case. Over 400 other P-ATPases are encoded in operons with a PacL1-like protein across bacteria, suggesting the broad conservation and function of metal efflux platforms.

## Results

### PacL1 is required for resistance to zinc intoxication in *M. tuberculosis*

PacL1 is encoded together with CtpC and contains a domain of unknown function (DUF), namely DUF1490 (Fig. 1a, http://pfam.xfam. org/family/PF07371). In silico analyses predict that PacL1, like most DUF1490-containing proteins, contains a transmembrane (TM) domain, a cytoplasmic part with Glu/Ala repeats, an intrinsically disordered region (IDR) of variable length, and a C-terminal metal-binding motif (MBM) (Fig. 1a). Moreover, PacL1 shows some homology to canonical chaperones, such as mycobacterial GroEL1 (Supplementary Fig. 1a). DUF1490-containing proteins form a family of previously undescribed proteins that are present exclusively in Actinobacteria. In

*M. tuberculosis*, the $P_{1B}$-ATPases CtpG and CtpV are also encoded with DUF1490-containing proteins, namely Rv1993c (PacL2) and Rv0968 (PacL3), respectively (Supplementary Fig. 1b); PacL1, PacL2 and PacL3 are highly similar at the primary sequence level and in their domain organization (Supplementary Fig. 1c).

To investigate whether PacL1 is involved in *M. tuberculosis* resistance to zinc poisoning, we generated mutants of *M. tuberculosis* lacking *pacL1* only, or the whole *pacL1-ctpC* operon (Supplementary Fig. 2a, b). RT-qPCR confirmed that *ctpC* was correctly transcribed in the Δ*pacL1* in-phase mutant (Supplementary Fig. 2c). In contrast to the wild-type strain, the Δ*pacL1* and Δ*pacL1-ctpC* mutants were not able to grow in liquid medium containing 100 μM $Zn^{2+}$ (Fig. 1b) nor to survive in the presence of 500 μM $Zn^{2+}$ (Fig. 1c). Complementation of the Δ*pacL1* mutant with *pacL1* restored resistance to zinc, whereas complementation of the Δ*pacL1-ctpC* mutant with both *pacL1* and *ctpC* was required to restore zinc resistance (Fig. 1b, c).

The non-pathogenic, fast-growing *M. tuberculosis* relative *Mycobacterium smegmatis* contains one homolog of the *pacL1-ctpC* operon, namely *msmeg_6059-msmeg_6058*. However, resistance to zinc in *M. smegmatis* was reported to be mediated by the cation diffusion facilitator MSMEG_0755, or ZitA[42]. We generated *M. smegmatis* mutants lacking *zitA* and/or the *msmeg_6059-msmeg_6058* operon (Supplementary Fig. 2d). The Δ*zitA* and Δ*zitA*Δ*msmeg_6059-msmeg_6058* (hereafter referred to as "Δ2") mutants were similarly sensitive to zinc concentrations above 50 μM; however the Δ*msmeg_6059-msmeg_6058* mutant was resistant to zinc, similar to the wild-type strain (Fig. 1d). These data confirmed that resistance to zinc intoxication in *M. smegmatis* is mediated exclusively by ZitA[42].

We generated recombinant Δ2 mutants expressing untagged or epitope-tagged versions of *M. tuberculosis* PacL1 and CtpC under the control of their native, zinc-inducible, promoter ($P_{pacL1}$) (Supplementary Fig. 2e). $P_{pacL1}$ was induced by $Zn^{2+}$ but not by $Mn^{2+}$ or $Cu^{2+}$, in *M. smegmatis* (Fig. 1e and Supplementary Fig. 2f) as in *M. tuberculosis* (12), and both PacL1 and CtpC proteins were correctly expressed in the recombinant strains (Supplementary Fig. 2g). Strikingly, expression of both PacL1 and CtpC in the Δ2 mutant fully restored resistance to zinc, whereas expression of either PacL1 or CtpC alone did not (Fig. 1f). Moreover, flow cytometry analysis of the strains incubated with the free zinc-specific dye FluoZin-3 revealed that the zinc-stressed Δ2 mutant accumulated zinc, and recombinant expression of both PacL1 and CtpC counteracted zinc accumulation (Fig. 1g). In contrast, expression of PacL1 and CtpC did not confer increased resistance to manganese intoxication to the recombinant *M. smegmatis* Δ2 mutant (Supplementary Fig. 2h). Altogether, these data show that the DUF1490 protein PacL1 is required for CtpC-mediated resistance to zinc excess, most likely through zinc efflux, in *M. tuberculosis*.

### PacL1 and CtpC cluster into membrane foci independent of flotillin

Next, we investigated the sub-cellular localization of CtpC and PacL1. As expected, recombinant His₆- or mVenus-tagged CtpC localized exclusively to the membrane fraction in *M. smegmatis* Δ2 co-expressing PacL1 (Fig. 2a and Supplementary Fig. 3a). Recombinant FLAG- or mTurquoise-tagged PacL1 also localized mostly to the membrane fraction in *M. smegmatis* Δ2 co-expressing CtpC (Fig. 2a and Supplementary Fig. 3a). Importantly, deletion of the TM domain in PacL1 (i.e., Leu⁷-Glu²⁵, resulting in "PacL1(ΔTM)") impaired this membrane localization of PacL1 (Fig. 2b) and complementation of the *M. tuberculosis* Δ*pacL1* mutant with this variant did not restore resistance to zinc excess (Fig. 2c). These data suggest that plasma membrane association is required for PacL1 function.

To investigate the structural conformation of PacL1, we purified a recombinant PacL1 that lacks the N-terminal 26 amino acids, encompassing the TM domain (soluble PacL1, "solPacL1", Supplementary Fig. 3b). Size-exclusion chromatography combined with multiangle

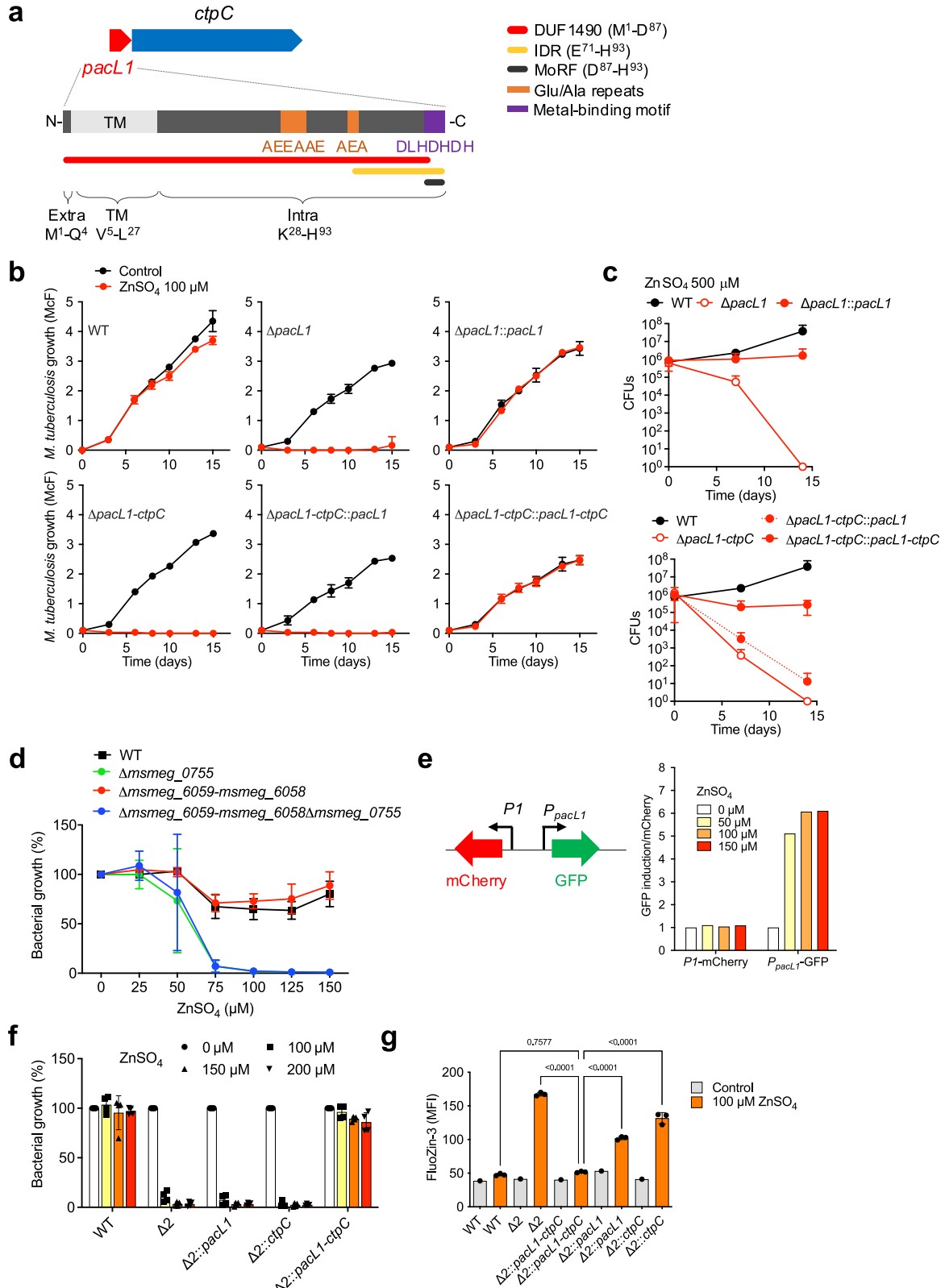

laser light scattering (SEC-MALS) analysis of solPacL1 revealed a radius of gyration of ~3 nm (Supplementary Fig. 3c), which is high for a ~7 kDa ($7238 \pm 351$ Da) protein, suggesting a largely unfolded conformation. These data are consistent with circular dichroism (CD) analysis (Supplementary Fig. 3d), and NMR $^1H_\alpha$ chemical shift index[43], which did not detect stable secondary structure elements (Supplementary Fig. 3e). Overall, our findings suggest that PacL1 is a membrane-associated

disordered protein, although we cannot exclude that PacL1 could adopt a folded conformation when associated with the plasma membrane.

We also visualized fluorescently tagged PacL1 and CtpC in live mycobacteria by epifluorescence microscopy. Surprisingly, mTurquoise-tagged PacL1 co-expressed with CtpC under the control of $P_{pacL1}$ clustered into dynamic patches in *M. smegmatis* Δ2 cells

**Fig. 1 | PacL1 is required for mycobacterial resistance to zinc intoxication.**
**a** Genetic organization of the *pacL1-ctpC* module, and polypeptide composition of DUF1490 protein Rv3269. Red tube indicates DUF1490. Yellow tube and black tube indicate intrinsically disordered region (IDR) and molecular recognition feature (MoRF), respectively, as predicted using $D^2P^2$ (ref. [76]). Protein topology was predicted using TMHMM Server v. 2.0. Extra extracellular part, TM transmembrane domain, Intra intracellular part. **b** *M. tuberculosis* H37Rv (wild-type, WT), *M. tuberculosis* mutants deleted in *pacL1* (Δ*pacL1*) or the *pacL1-ctpC* operon (Δ*pacL1-ctpC*), and their complemented strains (::*gene(s)*), were cultivated in 7H9 medium in the absence (Control, black lines) or presence of 100 μM ZnSO₄ (red lines) for 15 days. Bacterial growth was quantified by turbidity measurement (reported in McFarland's units). Data show mean ± s.d. of a biological replicate (*n* = 2) and are representative of 2 independent experiments. **c** Killing assay. Strains used in **a** were grown in 7H9 medium to OD₆₀₀ of 0.1. Next, bacteria were incubated in medium with 500 μM ZnSO₄ and bacterial viability was assessed by CFU scoring over time. Data show mean ± s.d of a biological replicate (*n* = 2), and are representative of 2 independent experiments. **d** *M. smegmatis* mc²155 (wild-type, WT), or *M. smegmatis* mutants inactivated in *msmeg_0755* (*zitA*), the *pacL1-ctpC* homologs *msmeg_6059-msmeg_6058*, or both modules, were cultivated in 7H9 medium supplemented with the indicated concentrations of ZnSO₄. Data

are expressed as % to control (0 ZnSO₄), show mean ± s.d. of a biological replicate (*n* = 3), and are representative of 2 independent experiments. **e** *M. smegmatis* Δ2 was transformed with an integrative vector encoding mCherry under the constitutive promoter *P1*, and GFP under the control of the native promoter of the *pacL1-ctpC* operon (*P_pacL1*). Bacteria were incubated overnight with the indicated concentrations of ZnSO₄, and GFP and mCherry signals (mean fluorescence intensity, MFI) were measured by flow cytometry. Data (*n* = 1) show GFP MFI/ mCherry MFI ratio, and are representative of 3 independent experiments. **f** *M. smegmatis* wild-type (WT), Δ2, or the Δ2 strain complemented with *pacL1*, *ctpC* or the *pacL1-ctpC* module were cultivated in 7H9 medium supplemented with the indicated concentrations of ZnSO₄ for 24 h. Bacterial growth was estimated by turbidity measurement, expressed as % of control (0 ZnSO₄), show mean ± s.d. of a biological replicate (*n* = 4), and are representative of 2 independent experiments. **g** The indicated strains were cultivated in 7H9 without ADC supplemented with 100 μM ZnSO₄ (orange) or no zinc (gray), incubated with FluoZin-3 at 1 mM for 1 h, and washed twice in PBS. Bacterial fluorescence was measured by flow cytometry and expressed as mean fluorescence intensity (MFI). Data show mean ± s.d of a biological replicate (*n* = 3), and are representative of 2 independent experiments. Data were analyzed using one-way ANOVA. *P* values are indicated. Source data are provided as a Source Data file.

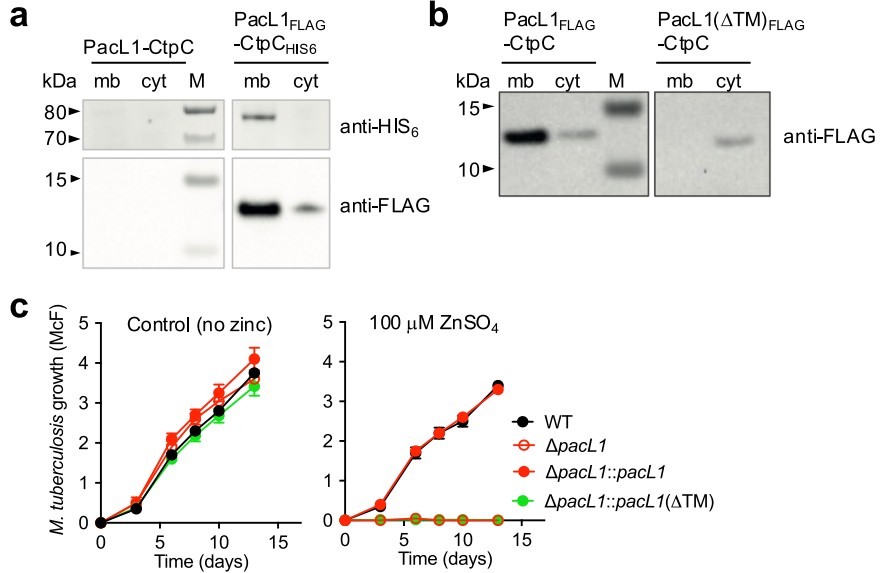

**Fig. 2 | PacL1 is a membrane-associated protein. a** Western blotting analysis of cytosolic (cyt) and membrane (mb) extracts of *M. smegmatis* Δ2 expressing recombinant native PacL1 and CtpC (left panel), or FLAG- and HIS₆-tagged PacL1 and CtpC, respectively (right panel). The top and bottom membranes were detected with anti-HIS₆ and anti-FLAG, respectively. M molecular size marker. Data are representative of 2 independent experiments. **b** Western blotting analysis of cytosolic (cyt) and membrane (mb) extracts of *M. smegmatis* Δ2 expressing recombinant CtpC with FLAG-tagged PacL1 (left panel), or FLAG-tagged PacL1 deleted of its transmembrane domain (ΔTM, right panel). The

membrane was treated for FLAG immuno-detection. M molecular size marker. Data are representative of 2 independent experiments. **c** *M. tuberculosis* H37Rv (wild-type, WT), the Δ*pacL1* mutant, or the Δ*pacL1* mutant complemented with *pacL1* or *pacL1*(ΔTM) were cultivated in complete 7H9 medium in the absence (left graph) or presence of 100 μM ZnSO₄ (right panel) for 14 days. Bacterial growth was evaluated by turbidity measurement (reported in McFarland's units). Data show mean ± s.d of a biological replicate (*n* = 3), and are representative of 2 independent experiments. Source data are provided as a Source Data file.

(Supplementary movie 1 and Fig. 3a), whereas mTurquoise-PacL1ΔTM exhibited only a faint signal evenly distributed in the bacterial cytoplasm (Fig. 3b). In addition to dynamic spots of PacL1, the majority of cells exhibited bright polar spots (Fig. 3a). We hypothesized that these spots corresponded to protein aggregates due to over-expression. In agreement, introduction of a down mutation in the *P_pacL1* promoter strongly reduced the polar spots, allowing precise counting of the PacL1 clusters per bacterial cell, with an average of 7 PacL1 spots per bacterial cell (Fig. 3c).

Live imaging of fluorescent CtpC also revealed dynamic spots in *M. smegmatis* Δ2 cells co-expressing fluorescent PacL1 (Supplementary movie 2). Importantly, we ascertained that fusion of PacL1 and CtpC to fluorescent proteins did not affect their function (Supplementary Fig. 4). However, the dynamic nature of the fluorescent spots and the

time lag between imaging of the two proteins precluded us from quantifying their potential co-localization. Therefore, we examined formaldehyde-fixed *M. tuberculosis* expressing fluorescent PacL1 and CtpC, which revealed strong co-localization of the two proteins (Fig. 3e), with a Pearson's correlation coefficient of 0.8 ± 0.1 (Fig. 3f).

The clustered distribution of PacL1 and CtpC in the plasma membrane resembled functional membrane microdomains (FMMs) reported in several bacterial species[40,41,44–50]. Assembly of such FMMs is mediated by flotillin-like scaffold proteins, e.g., FloT and FloA in *Bacillus subtilis*[48]. Although their role as "universal" scaffold proteins for common FMMs remains a matter of discussion[51,52], flotillins co-localize, at least partially, with several membrane proteins, favor interactions between protein involved in various biological processes, and their inactivation results in partial or complete destabilization of

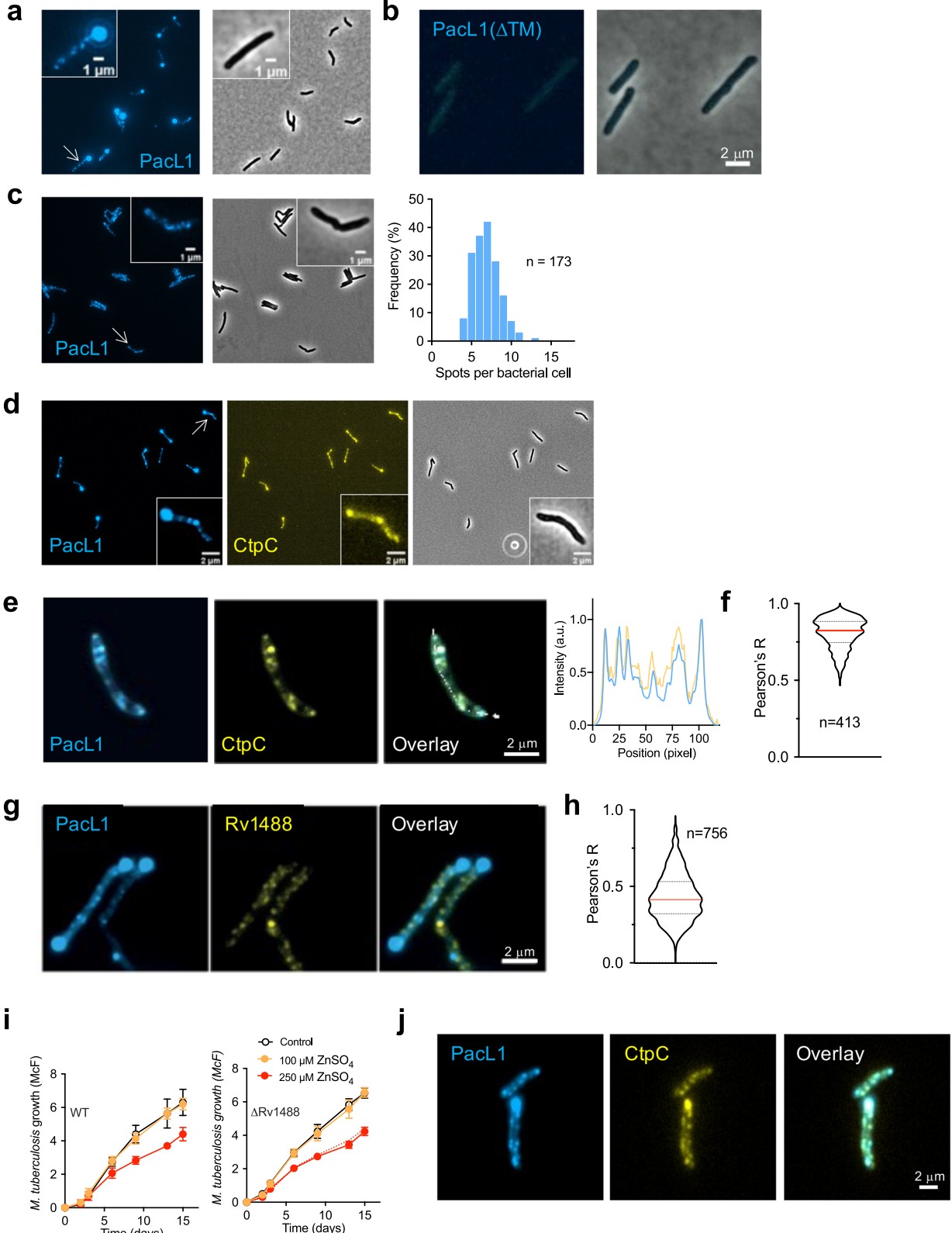

these proteins in the plasma membrane[41]. Therefore, we investigated whether flotillins are required to assemble PacL1 and CtpC at the plasma membrane in *M. tuberculosis*. BLAST analysis revealed a single FloT homolog, namely Rv1488 (27% identity, 53% similarity), and no FloA homolog, in *M. tuberculosis*. Like FloT, Rv1488 contains a putative membrane-associated region (MAR), a band-7/prohibitin (PHB) domain (Supplementary Fig. 5a) and several Glu/Ala repeats. Rv1488

assembles into dynamic patches in the mycobacterial plasma membrane (Supplementary Fig. 5b, c and Supplementary movie 3). However, mVenus-Rv1488 did not substantially co-localize with mTurquoise-PacL1 (Fig. 3g), with a Pearson's correlation coefficient of $0.4 \pm 0.1$ (Fig. 3h). In addition, in a triple fluorescence system, mCherry-Rv1488 formed patches distinct from those of mTurquoise-PacL1 and mVenus-CtpC (Supplementary Fig. 5d). Moreover, a

**Fig. 3 | PacL1 and CtpC cluster into dynamic membrane foci in a flotillin-independent manner.** *M. smegmatis* expressing mTurquoise-tagged PacL1 (**a**, **c**) or mTurquoise-tagged PacL1ΔTM (**b**) under the control of the native $P_{pacL1}$ promoter (**a**, **b**) or the "down" $P_{pacL1}$ promoter, in which T$^{-61}$G$^{-60}$ were mutated into CC to reduce transcription (**c**), was examined using an Eclipse TI-E/B wide field epifluorescence microscope. The right panel in **c** shows the distribution of spot number/bacterial cell, counted by visual examination in $n = 173$ cells. **d** Epifluorescence microscopy image of *M. smegmatis* expressing mTurquoise- and mVenus-tagged PacL1 and CtpC, respectively. The proteins were expressed under the control of the $P_{pacL1}$ promoter and bacteria were grown in complete 7H9 medium. **e** Formaldehyde-fixed *M. tuberculosis* expressing mTurquoise- and mVenus-tagged PacL1 and CtpC, respectively, was examined using an Eclipse TI-E/B wide field epifluorescence microscope. Right graph shows co-localization along the white line in the overlay panel. The proteins were expressed under the control of the $P_{pacL1}$ promoter and bacteria were grown in complete 7H9 medium. **f** Co-localization index between blue (PacL1) and yellow (CtpC) pixels in $n = 413$ cells. **g** *M. smegmatis* expressing mTurquoise- and mVenus-tagged PacL1 and Rv1488/FloT, respectively, was examined using an Eclipse TI-E/B wide field epifluorescence microscope. **h** Co-localization index between blue (PacL1) and yellow (Rv1488) pixels in $n = 756$ cells. **i** *M. tuberculosis* H37Rv (wild-type, WT, left panel), or a *M. tuberculosis* mutant deleted in Rv1488 (ΔRv1488) were cultivated in complete 7H9 medium in the absence (Control, black lines) or presence of 100 μM or 250 μM ZnSO$_4$ for 15 days. Bacterial growth was quantified by turbidity measurement (reported in McFarland's units). Data show mean ± s.d of a biological replicate ($n = 3$), and are representative of 2 independent experiments. In the right panel, the dotted line reproduces data for WT in the left panel to show that there is no difference between the WT and the ΔRv1488 mutant in sensitivity to zinc at 250 μM. **j** Formaldehyde-fixed *M. tuberculosis* mutant ΔRv1488 expressing mTurquoise- and mVenus-tagged PacL1 and CtpC, respectively, was examined using an Eclipse TI-E/B wide field epifluorescence microscope. Source data are provided as a Source Data file.

ΔRv1488 *M. tuberculosis* mutant (Supplementary Fig. 5e) was resistant to zinc intoxication, similar to the parental wild-type strain (Fig. 3i) and showed proper clustering of PacL1 and CtpC (Fig. 3j). Altogether, these results suggest that PacL1 and CtpC cluster into foci in the mycobacterial plasma membrane independent of flotillin.

## PacL1 has zinc-binding site required for resistance to high zinc concentrations

PacL1 contains a putative C-terminal MBM, D$^{87}$LHDHDH$^{93}$, in which the underlined His residues are predicted to bind zinc[53]. We generated and purified solPacL1Δ3, a recombinant version of solPacL1 lacking the last three residues of the MBM (Supplementary Fig. 3b–d). SEC-MALS analysis indicated that like solPacL1, solPacL1Δ3 is monodisperse and displays a calculated molecular mass of 6,764 ± 430 Da (Supplementary Fig. 3c). CD analysis showed that like solPacL1, solPacL1Δ3 has an overall disordered structure (Supplementary Fig. 3d).

Dialysis at equilibrium analysis revealed that solPacL1 bound zinc at a 1:1 molar ratio, whereas solPacL1Δ3 did not bind zinc (Fig. 4a). The apparent $K_D$ of solPacL1 for zinc was of 8.6 ± 6.4 μM. Isothermal titration calorimetry (ITC) confirmed this finding (Fig. 4b), with an apparent $K_D$ of ~3 μM. Of note, solPacL1 did not bind manganese (Fig. 4a). Furthermore, NMR analysis of solPacL1 in the absence or presence of 2 equivalents of zinc showed that the resonances of residues at the C-terminus, encompassing D$^{87}$LHDHDH$^{93}$, were broadened beyond detection due to chemical exchange on an intermediate (μs-ms) time scale (Fig. 4c). The chemical shift perturbations of amide protons decreased progressively from the metal-binding site toward the N-terminus of solPAcL1 (Supplementary Fig. 6). Finally, analysis of a synthetic Trp-MBM peptide (W-DLHDHDH) revealed binding to zinc at a 1:1 molar ratio, with an apparent $K_D$ of ~6 μM (Supplementary Fig. 7a–d), and no binding to manganese (Supplementary Fig. 7e).

To assess the function of the PacL1 MBM in vivo, recombinant *M. tuberculosis* Δ*pacL1* expressing CtpC together with PacL1Δ3 (lacking the H$^{91}$DH$^{93}$ residues) or "PacL1Δ7" (lacking the D$^{87}$LHDHDH$^{93}$ residues) were cultivated in the presence of zinc at concentrations above the toxic threshold. Intriguingly, the Δ*pacL1* mutants expressing CtpC and PacL1Δ3 or PacL1Δ7 showed resistance to zinc at 100 μM but not at 250 μM (Fig. 4d). Indeed, PacL1Δ3 and PacL1Δ7 restored growth in 100 μM zinc as well as wild type PacL1.

Overall, these results show that PacL1Δ7 is sufficient for resistance to 100 μM zinc, but that the PacL1 MBM is required for mycobacterial resistance to zinc at higher concentrations. Thus, unlike previously reported copper metallochaperones[39], PacL1 plays MBM-dependent and MBM-independent roles in zinc resistance in *M. tuberculosis*.

## PacL1 is a chaperone-like protein required for abundant CtpC levels

PacL1 shows similarity to canonical chaperones, such as GroEL1 (Supplementary Fig. 1a), and contains Glu/Ala repeats (Supplementary Fig. 1c) known to promote flotillin-mediated protein-protein interactions and protein stability in FMMs[40,54].

To investigate whether the Glu/Ala repeats in PacL1 were required to assemble CtpC into membrane foci, we generated ΔA$^{54}$-S$^{86}$ and ΔR$^{37}$-S$^{86}$, truncated versions of PacL1 which still contain the MBM but lack the Glu/Ala repeats (Supplementary Fig. 8a). PacL1ΔA$^{54}$-S$^{86}$ and PacL1ΔR$^{37}$-S$^{86}$ were both highly expressed and localized to the plasma membrane in *M. smegmatis* Δ2, comparable to the native protein (Fig. 5a and Supplementary Fig. 8a and 8b). CtpC co-expressed with the PacL1Δ7 variant lacking the C-terminal MBM was still highly abundant in the membrane (Fig. 5a). Intriguingly, however, CtpC levels were severely diminished when co-expressed with the PacL1 ΔA$^{54}$-S$^{86}$ or ΔR$^{37}$-S$^{86}$ variants compared to native Pac L1 (Fig. 5a; Supplementary Fig. 8a, b) and, in agreement, *M. smegmatis* Δ2 co-expressing CtpC with the ΔA$^{54}$-S$^{86}$ or ΔR$^{37}$-S$^{86}$ PacL1 variants remained sensitive to 100 μM zinc (Fig. 5b). RT-qPCR analysis confirmed that *ctpC* transcript levels were similar in all complemented conditions (Supplementary Fig. 8c).

We generated a *M. smegmatis* Δ2 strain expressing $P_{pacL1}$-driven CtpC and anhydrotetracycline (Atc)-inducible PacL1 (Fig. 5c). Addition of Atc resulted in accumulation in the plasma membrane not only of PacL1 but also of CtpC (Fig. 5c), and restored resistance to zinc (Fig. 5d). Altogether, these results revealed that the cytoplasmic region of PacL1 is required for abundant CtpC levels, and for proper function of PacL1 triggering resistance to zinc excess.

To further investigate the specific contribution of the Glu/Ala repeats, we generated Glu$^{55}$ > Ala, Glu$^{59}$ > Val and Glu$^{71}$ > Ala single substitution mutants of PacL1, and a triple mutant in Glu$^{55}$, Glu$^{59}$ and Glu$^{71}$, hereafter called "3EA". Whereas Δ2 expressing PacL1 variants with the single Glu substitutions restored resistance to 100 μM zinc, those expressing 3EA were as sensitive as parental Δ2 (Fig. 6a) and accumulated intracellular zinc (Fig. 6b). In addition, CtpC levels were severely reduced in the Δ2 strain expressing the PacL1 3EA variant (Fig. 6c), compared to the wild-type PacL1 or the single substitution PacL1 E$^{71}$A. RT-qPCR analysis confirmed that *pacL1* and *ctpC* were correctly transcribed in these recombinant strains (Supplementary Fig. 8d). Additionally, mVenus-CtpC levels were strongly reduced in *M. smegmatis* Δ2 expressing mTurquoise-PacL1(3EA) *vs.* mTurquoise-PacL1 (Supplementary Fig. 8e).

Altogether, these results show that the Glu/Ala repeats of PacL1 are required for high levels of membrane-bound P-ATPase CtpC and for efficient function of the PacL1/CtpC zinc efflux system. In addition, this function is independent of the PacL1 C-terminal MBM. Based on these data we propose that PacL1 resembles bacterial flotillin-like proteins, which act as scaffolds to promote stability of membrane-associated proteins. Supporting this hypothesis, bipartite split-GFP experiments indicated that PacL1 can physically interact with the CtpC N-terminal end, and that this interaction requires the E/A repeats (Fig. 6d and Supplementary Fig. 9). Furthermore, Blue-native polyacrylamide gel electrophoresis indicated that both CtpC and PacL1 are

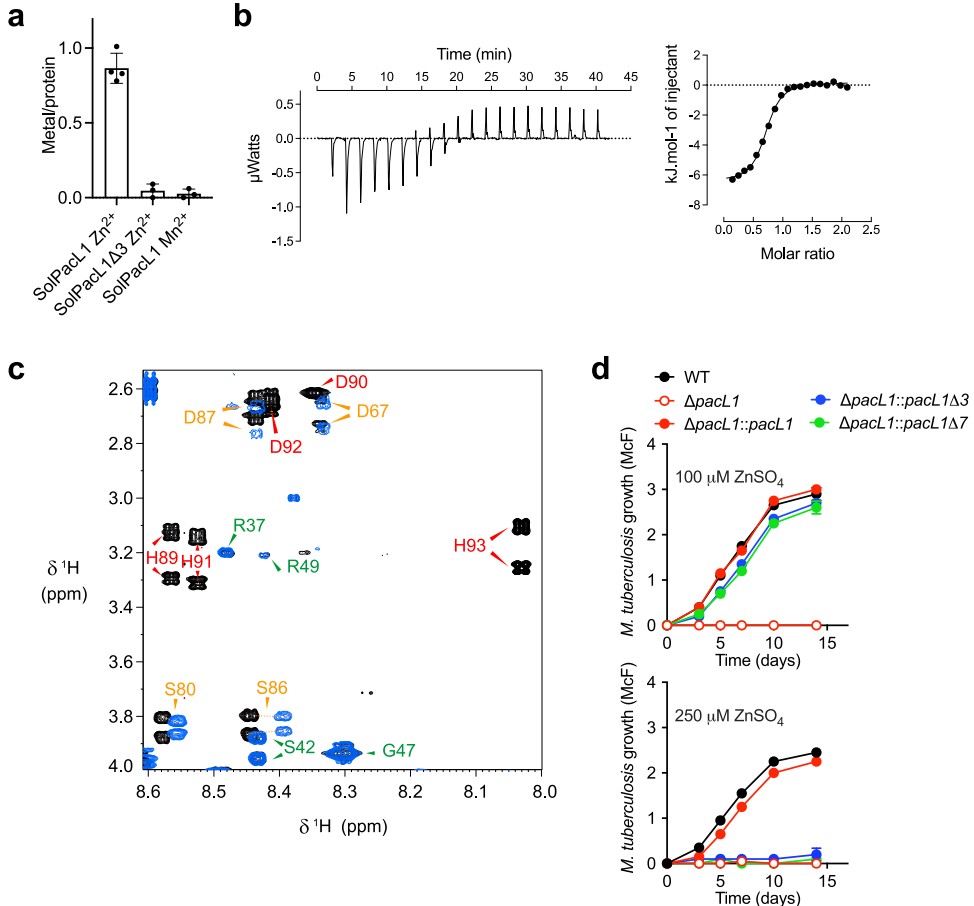

**Fig. 4 | PacL1 binds zinc, but not manganese, in its C-terminal metal-binding motif. a** Measurement of metal binding to solPacL1 and solPacL1Δ3 (50 μM) by dialysis at equilibrium. Experiment was performed in 50 mM MOPS pH 7, 200 mM NaCl during 21 h at 4 °C in the presence of 50 μM of the indicated metal. Data show mean ± s.d. of three independent experiments (*n* = 4 per experiment). **b** Isothermal titration calorimetry data. Thermograms and corresponding titration curves obtained by successive injections of 2 μL of 1.5 mM Zn[CH₃COO]₂ into the sample containing 150 μM of purified solPacL1. Molar ratio is defined as the number Zn per solPacL1. The data are representative of two independent experiments. **c** Superposition of 2D NMR TOCSY spectra (80 ms mixing time) of solPacL1 (500 μM in 20 mM MES-d₁₃ pH 6.0, 100 mM NaCl at 7 °C) in the absence (black) and presence (blue) of 2 equivalents of zinc acetate. For clarity, the

region of the spectrum containing $H_N$-$H_β$ cross-peaks of histidines in the HDHDH zinc-binding motif is shown. The assignment is reported using the sequence numbering and colors represent the strength of perturbation induced by zinc binding; red: strong (broadened beyond detection); orange: medium; green: no effect. **d** *M. tuberculosis* H37Rv (wild-type, WT), the Δ*pacL1* mutant, or the Δ*pacL1* mutant complemented with *pacL1*, *pacL1Δ3* (lacking the H⁹¹DH⁹³-encoding sequence) or *pacL1Δ7* (lacking the D⁸⁷LHDHDH⁹³-encoding sequence) were cultivated in complete 7H9 medium in the presence of 100 μM (upper panel) or 250 μM (lower graph) ZnSO₄ for 14 days. Bacterial growth was evaluated by turbidity measurement (reported in McFarland's units). Data show mean ± s.d of a biological replicate (*n* = 2), and are representative of 2 independent experiments. Source data are provided as a Source Data file.

found in high-molecular weight complexes at the plasma membrane (Fig. 6e).

### *M. tuberculosis* PacL proteins cluster together at the membrane and exhibit partially redundant functions

Finally, we investigated whether PacL2 and PacL3, encoded with different P-ATPases in distinct operons with different regulations, have overlapping localization and function with PacL1. Fixed mycobacteria expressing fluorescently labeled PacL proteins showed colocalization of PacL1, PacL2 and PacL3 in common clusters in the mycobacterial plasma membrane, with Pearson's correlation coefficients of 0.8 ± 0.1 (PacL1/PacL2) and 1.0 ± 0.02 (PacL1/PacL3) (Fig. 7a, b).

We generated a triple *pacL1*-, *pacL2* and *pacL3*-deficient mutant of *M. tuberculosis*[55], together with several complemented strains expressing target genes from the zinc-responsive promoter $P_{pacL1}$. As expected, the triple mutant was sensitive to zinc at 100 μM or 250 μM, and zinc resistance was restored in a strain complemented with $P_{pacL1}$::PacL1 (Fig. 7c, upper panels). Strikingly, complementation with $P_{pacL1}$-driven CtpC and PacL3, which is cognate to CtpV and contains a C-terminal putative metal-binding motif (D⁹³DGHDH⁹⁸), restored

resistance to zinc at 100 μM, and partially restored resistance to zinc at 250 μM (Fig. 7c, lower left panel). Complementation with $P_{pacL1}$-driven CtpC and PacL2, which is cognate to CtpG and contains a C-terminal putative metal-binding semi-motif only (D⁸⁹E⁹⁰), partially restored resistance to zinc at 100 μM, but resistance to zinc at 250 μM was poor (Fig. 7c, lower middle panel). Finally, complementation of the triple mutant with a plasmid encoding CtpC and a PacL2-MBM fusion protein, containing the PacL1 zinc-binding motif, phenocopied the complemented mutant expressing CtpC and PacL1 (Fig. 7c, lower right panel). Similarly, complementation of a *pacL1*-deficient single mutant with $P_{pacL1}$-driven PacL2 or PacL3 completely or partially restored resistance to zinc excess at 100 μM ZnSO₄, respectively, but not at 250 μM ZnSO₄, further demonstrating that the PacL1 MBM is required for resistance to zinc at high concentrations (Supplementary Fig. 10).

The experiments above were performed using strains expressing PacL1, PacL2 or PacL3 under the control of the zinc-inducible $P_{pacL1}$ promoter. In *M. tuberculosis*, PacL2 and PacL3 are not induced by zinc; specifically, the *pacL2-ctpG* module is induced by cadmium and lead[56], and the *pacL3-ctpV* module is induced by copper[57]. Thus, we assessed whether addition of Cd²⁺ to a Zn²⁺-stressed Δ*pacL1 M. tuberculosis*

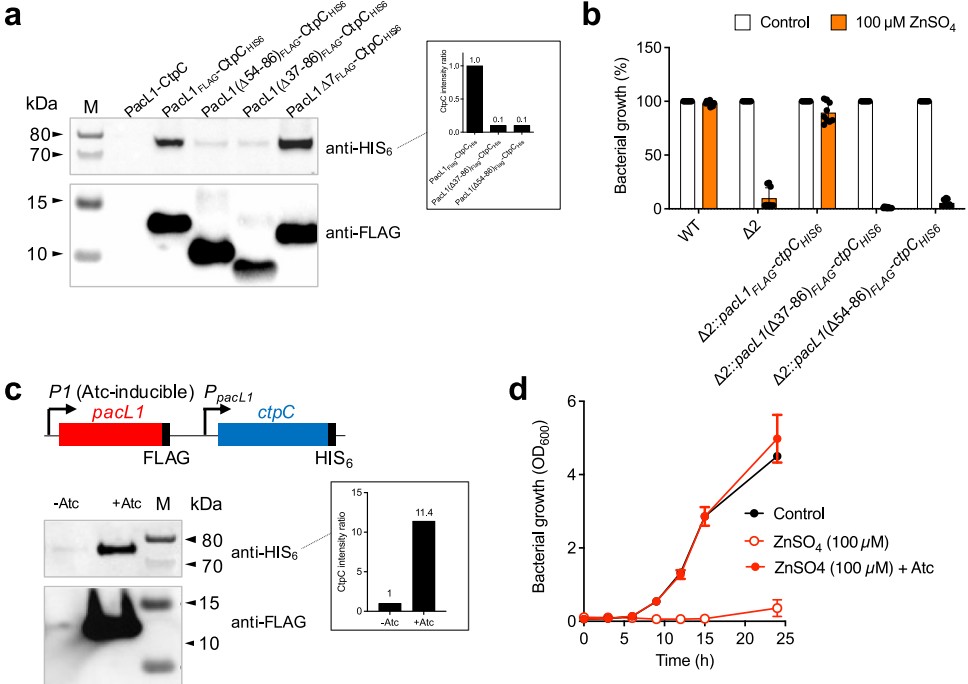

**Fig. 5 | PacL1 is required for abundant CtpC levels and resistance to zinc intoxication. a** Western-blotting analysis of membrane extracts from *M. smegmatis* Δ2 expressing PacL1 and CtpC, or a combination of HIS$_6$-tagged CtpC and FLAG-tagged PacL1, FLAG-tagged PacL1ΔR$^{37}$S$^{86}$, containing the MBM, FLAG-tagged PacL1ΔA$^{54}$S$^{86}$, containing the MBM, or FLAG-tagged PacL1Δ7, lacking the MBM. All genes are expressed under the control of the native P$_{PacL1}$ promoter and bacteria were grown in 7H9 medium. The upper membrane was treated for HIS$_6$ immuno-detection and the lower membrane was treated for FLAG immuno-detection. M molecular size marker. Relative CtpC-HIS$_6$ protein quantification is displayed in the inset. Data show quantification of one WB (displayed on the left), and are representative of two independent experiments. **b** Strains used in **a** were cultivated for 24 h in complete 7H9 medium in the absence (Control, white bars) or presence (orange bars) of 100 µM ZnSO$_4$. Bacterial growth was quantified by turbidity measurement (expressed as % of untreated control).

Data show mean ± s.d of a biological replicate ($n$ = 3), and are representative of 2 independent experiments. **c** Western-blotting analysis of membrane extracts from *M. smegmatis* Δ2 expressing Atc-inducible FLAG-tagged PacL1 and constitutive HIS$_6$-tagged CtpC (left panel). The membrane was cut into two pieces: the upper part was treated for HIS$_6$ immuno-detection and the lower part was treated for FLAG immuno-detection. M molecular size marker. Data are representative of 2 independent experiments. CtpC-HIS$_6$ protein quantification is displayed in the inset (relative to amount in the non-Atc-induced strain). **d** The strain used in **c** was cultivated in complete 7H9 medium in the absence (Control, black circles) or presence (red circles) of 100 µM ZnSO$_4$, without (open circles) or with (closed circles) Atc. Bacterial growth was quantified by turbidity measurement (expressed in OD$_{600}$ units). Data show mean ± s.d of a biological replicate ($n$ = 2), and are representative of 2 independent experiments. Source data are provided as a Source Data file.

mutant might restore resistance to zinc by inducing *pacL2*. As expected, the Δ*pacL1* mutant was highly sensitive to zinc at 100 µM; however, the addition of 5 µM Cd$^{2+}$ was sufficient to partially restore resistance to zinc in this mutant (Fig. 7d), phenocopying the complemented triple *pacL* mutant expressing CtpC and PacL2. Addition of Cd$^{2+}$ to a mutant lacking both PacL1 and CtpC did not restore zinc resistance in the mutant (Fig. 7d, right panel), further demonstrating that CtpC is essential for resistance to zinc intoxication, even if the function of PacL1 can be fulfilled, at least partially, by PacL2. By contrast, we did not observe rescue of zinc-stressed Δ*pacL1 M. tuberculosis* by PacL3 induction upon the addition of 5 or 50 µM CuSO$_4$ (data not shown). Unlike cadmium, copper can exist in two oxidation states, which can vary depending on the experimental conditions. This might explain this unexpected finding, and would need further investigation.

Altogether, these results show that PacL proteins colocalize in the mycobacterial plasma membrane, and that these proteins fulfill partially redundant functions in P-ATPase-mediated resistance to metal intoxication, depending on the metal microenvironment.

## Discussion

Microbial pathogens navigate different microenvironments with various metal contents during infection and metal efflux systems are required by various microbes for survival inside host phagocytes[12,13,17–21]. In particular, the P-ATPase CtpC is required for the intracellular pathogen *M. tuberculosis* to resist zinc intoxication and

multiply inside host macrophages[12]. The present study reveals that the DUF1490 protein PacL1 is necessary for mycobacterial resistance to excess zinc, and plays an unexpected dual role: 1) The MBM of PacL1 is required for resistance to high zinc concentrations, suggesting that PacL1 might act as a zinc scavenger or zinc metallochaperone, or both, and 2) its cytoplasmic region is required for abundant CtpC levels and resistance to lower zinc concentrations (Supplementary Fig. 11).

Unexpectedly, and in contrast to our previous findings[12], single *pacL1*-KO and triple *pacL1-3*-KO mutants did not display growth defect in human macrophages (data not shown). One explanation for this apparent discrepancy might be that *M. tuberculosis* GC1237 (lineage 2) was used in Botella et al., while the less virulent H37Rv strain (lineage 4) was used in the present study. It is possible that H37Rv induces a reduced zinc burst, compared to GC1237, in macrophages. In addition, during the revision of this manuscript Mehdiratta et al. reported the identification of novel zinc metallophores, called kupyaphores, secreted by *M. tuberculosis*[58]. The authors showed that kupyaphores are involved in both uptake of zinc in zinc-limited conditions and in resistance to zinc excess. The mechanism by which kupyaphores mediate resistance to zinc excess is not clear; yet a kupyaphore-deficient mutant was found to have attenuated virulence in vivo in mice. How kupyaphores and CtpC work together to sustain mycobacterial resistance to zinc excess will need to be investigated; it is possible that *ctpC* deficiency in a kupyaphore-deficient background might reveal further attenuation.

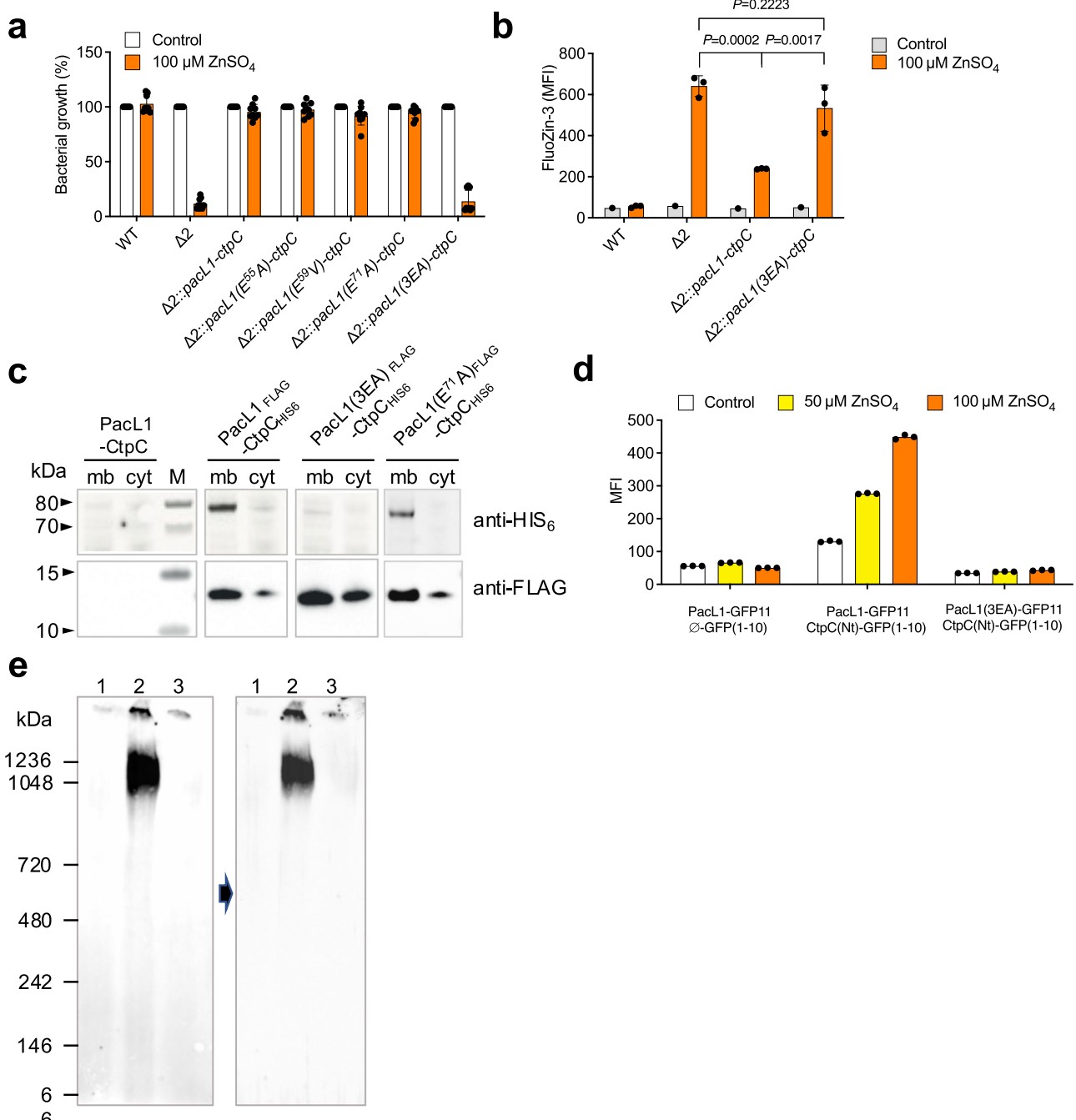

**Fig. 6 | PacL1 Glu/Ala repeats are required for abundant CtpC levels, resistance to zinc intoxication and formation of high molecular weight complexes in the mycobacterial membrane. a** *M. smegmatis* wild-type strain (WT), Δ2 mutant, or Δ2 expressing recombinant CtpC and native PacL1 or PacL1 $E^{55}A$, $E^{59}V$, $E^{71}A$ or 3EA variants were cultivated for 24 h in complete 7H9 medium in the absence (Control, white bars) or presence (orange bars) of 100 μM $ZnSO_4$. Bacterial growth was quantified by turbidity measurement (expressed as % of untreated control). Data show mean ± s.d of a biological replicate (*n* = 3), and are representative of 2 independent experiments. **b** The indicated *M. smegmatis* strains were cultivated in complete 7H9 without ADC supplemented with 100 μM $ZnSO_4$, incubated with FluoZin-3 at 1 mM for 1 h, and washed twice in PBS. Bacterial fluorescence was measured by flow cytometry and expressed as mean fluorescence intensity (MFI). Data show mean ± s.d of a biological replicate (*n* = 3), and are representative of 2 independent experiments. Data were analyzed using one-way ANOVA. **c** Western-blotting analysis of cytosolic (cyt) and membrane (mb) fractions from *M. smegmatis* Δ2 expressing recombinant PacL1 and CtpC, or $HIS_6$-tagged CtpC and FLAG-tagged PacL1, or the PacL1 $E^{71}A$ or 3EA variants. The upper membrane was treated for $HIS_6$ immuno-detection and the

lower membrane was treated for FLAG immune-detection. Data are representative of 2 independent experiments. **d** Bipartite split-GFP experiment using *M. smegmatis* Δ2 expressing a PacL1-GFP11 fusion protein and control GFP1-10 (left), a PacL1-GFP11 fusion protein and the CtpC N-terminal region ($M^1$-$A^{85}$) fused to GFP1-10 (middle), or a PacL1(3EA)-GFP11 fusion protein and the CtpC N-terminal region ($M^1$-$A^{85}$) fused to GFP1-10 (right). Bacteria were grown in complete 7H9 medium supplemented with the indicated amounts of zinc. Fluorescence was recorded by flow cytometry and expressed as mean fluorescent intensity (MFI). Data (*n* = 3 technical replicates) are representative of 2 independent experiments. **e** Blue-native gel analysis of membrane proteins from *M. smegmatis* Δ2 expressing recombinant PacL1 and CtpC (lanes 1), $HIS_6$-tagged CtpC and FLAG-tagged PacL1 (lanes 2), or $HIS_6$-tagged CtpC and FLAG-tagged PacL1(3EA) variant (lanes 3). Proteins were extracted from *M. smegmatis* membrane fractions with 0.25% DDM, separated on native gel, transferred onto PVDF membranes and revealed by FLAG (left panel) or $HIS_6$ (right panel) immune-detection. Black arrow indicates membrane stripping. Data are representative of two independent experiments. Reverse immune-detection ($HIS_6$ followed by FLAG) led to similar results. Source data are provided as a Source Data file.

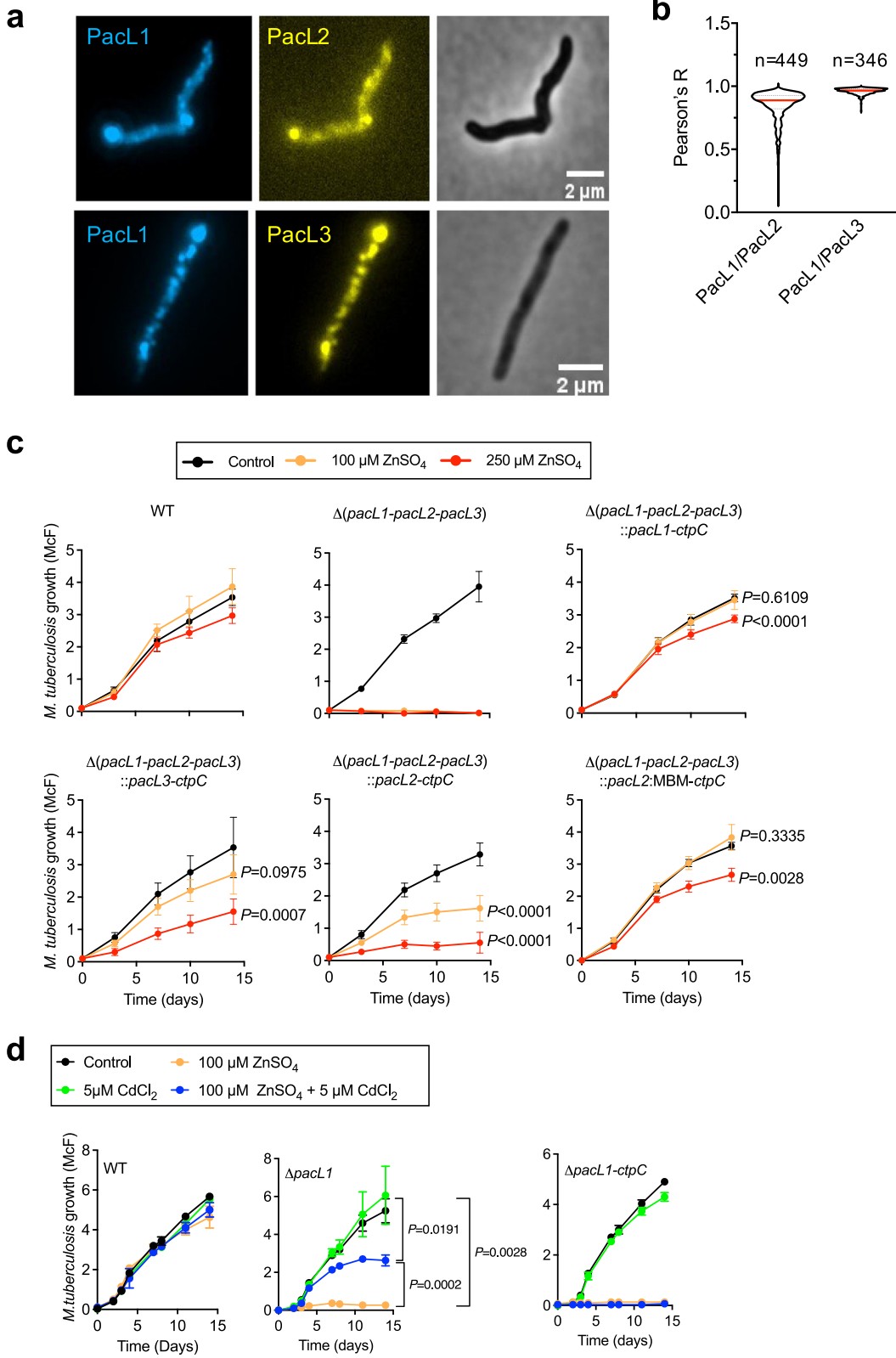

CtpC had been proposed to transport manganese, rather than zinc[35]. Yet, the velocity of metal transport by CtpC found in that study was unusually low, with 1.2 $Mn^{2+}$ and 0.4 $Zn^{2+}$ ions transported per minute on average, which is dramatically less than for other previously characterized P-ATPases. For example, *Escherichia coli* ZntA transports 20–45 $Zn^{2+}$ ions per minute[59,60] and *Thermotoga maritima* CopA transports over 140 $Cu^+$ ions per minute[61]. Given the ability of PacL1 to

bind zinc, but not manganese, and even more importantly that PacL1 is absolutely required for CtpC stabilization in the plasma membrane, it is most likely that co-expression of PacL1 and CtpC increases CtpC metal specificity and velocity, which will need further investigation. Unfortunately, and despite extensive attempts, we have been unable to produce CtpC, alone or together with PacL1, in sufficient quantity and quality in various recombinant systems, namely *E. coli*, *Saccharomyces*

**Fig. 7 | PacL1, PacL2 and PacL3 co-localize in shared membrane foci and partially fulfill similar functions. a** Epifluorescence microscopy examination of fixed *M. smegmatis* expressing mTurquoise-tagged PacL1 and mVenus-tagged PacL2 (upper panels) or -PacL3 (lower panels). All tagged proteins were expressed under the control of the $P_{pacL1}$ promoter, and bacteria were grown in complete 7H9 medium. Images are representatives of 3 independent experiments. **b** Co-localization index between PacL1 and PacL2, and between PacL1 and PacL3. *n* indicates the number of counted cells. All tagged proteins were expressed under the control of the $P_{pacL1}$ promoter, and bacteria were grown in complete 7H9 medium. **c** *M. tuberculosis* wild-type (WT), a *M. tuberculosis* triple mutant inactivated in *pacL1-3* (Δ*pacL1-pacL2-pacL3*) or the triple mutant complemented with vectors encoding PacL1 and CtpC, PacL3 and CtpC, PacL2 and CtpC or a PacL2:MBM fusion protein and CtpC, were cultivated in complete 7H9 medium supplemented with ZnSO₄ at the indicated concentrations (Control, no zinc added). Bacterial growth was quantified by turbidity measurement (expressed in McFarland's units). Data show mean ± s.d of a biological replicate (*n* = 6), and are representative of 2 independent experiments. Data at day 14 were analyzed by two-sided *t*-test after passing the Shapiro–Wilk and/or Kolmogorov–Smirnov normality tests. *P* values are given compared to untreated control. **d** *M. tuberculosis* wild-type (WT) and *M. tuberculosis* Δ*pacL1* or Δ*pacL1-ctpC* mutants were cultivated in complete 7H9 medium supplemented with 100 μM ZnSO₄, 5 μM CdCl₂, 100 μM ZnSO₄ and 5 μM CdCl₂, or no metal (Control). Bacterial growth was assessed by turbidity measurement as expressed in McFarland's units. Data show mean ± s.d of a biological replicate (*n* = 3), and are representative of 2 independent experiments. Data at day 14 were analyzed by two-sided *t*-test after passing the Shapiro–Wilk and/or Kolmogorov–Smirnov normality tests. Source data are provided as a Source Data file.

*cerevisiae*, and *M. smegmatis*, to perform functional studies (data not shown).

How metallochaperones work together with P-ATPases is still unclear. The dominant paradigm is that the N-terminal P-ATPase MBD plays a regulatory role, rather than being involved in direct metal transfer to the transmembrane metal-binding site of P-ATPases[16,26,62]. Because the CtpC N-terminal MBD is essential for mycobacterial resistance to zinc intoxication (data not shown), we hypothesize that PacL1 is involved in metal donation to the MBD, as was reported for copper metallochaperones and Cu⁺-ATPases[36–38]. However, PacL1 and DUF1490 proteins are fundamentally different from copper metallochaperones, which are typically well-structured and exhibit a βαββαβ ferredoxin-like fold[36–38]. DUF1490 proteins might therefore constitute an unprecedented class of P-ATPase-associated metallochaperones. In support of this hypothesis, the line broadening observed by NMR in the presence of zinc may arise from metal ion exchange between different coordination conformers with the DLHDHDH motif. Such line-broadening was reported for $ZiaA_N$, the soluble N-terminal MBD of a zinc transporting ATPase in *Synechocystis*[63]. $ZiaA_N$ contains two zinc-binding domains, and the authors suggested that dynamic interactions between the atypical XH7 binding motif at the unstructured C-terminus and the canonical CXXC binding site of the ferredoxin-fold region might be involved in the transport mechanism[63]. PacL1 shows some similarity to the region of ZiaA encompassing the unfolded XH7 MBM and the first transmembrane helix, suggesting an analogous mechanism in which the flexible cytosolic domain of PacL1, anchored in the membrane in proximity to CtpC, binds metal cations and transfers them to the N-terminal MBD of CtpC.

Importantly, unlike any other metallochaperones, PacL1 is required for abundant levels of its cognate P-ATPase. We propose that PacL1 stabilizes and promotes membrane incorporation of CtpC in metal efflux platforms. We showed that PacL1/CtpC foci exclude the flotillin-like FloT (Rv1488), and that mycobacterial FloT is not required for *M. tuberculosis* resistance to zinc stress. However, like FloT, PacL1 contains Glu/Ala repeats that are required for stabilization of CtpC into metal efflux platforms, suggesting PacL1 might act, like bacterial flotillins, as a scaffold protein and as a molecular chaperone. Our split-GFP and Blue-native PAGE data indicate that PacL1 can interact with CtpC and that the two proteins form high molecular weight complexes in the plasma membrane. Whether and how this interaction would sustain CtpC stabilization and/or metal transfer from PacL1 to CtpC will need further investigation. Equally important will be to investigate the molecular lipid and protein composition of the PacL/P-ATPase-containing metal efflux platforms in order to explore whether our metal efflux platforms represent bona fide "microdomains", i.e., incorporating specific membrane lipids. This could be achieved using proximity-dependent biotinylation[64] assays and the recently developed styrene-maleic acid anhydride lipid particle (SMALP) technology[65] followed by proteomics and lipidomics analyses.

It will be of interest to evaluate whether PacL2 and PacL3 function in a similar manner with their cognate P-ATPase, CtpG and CtpV,

respectively. PacL2 contains a C-terminal putative metal-binding semi-motif only (D⁸⁹E⁹⁰); it is possible that PacL2 needs to oligomerize to bind metal ions. PacL3 is even more striking since its cognate P-ATPase, CtpV, is thought to mediate copper efflux[20], yet PacL3 does not contain any apparent copper-binding residues but rather a putative zinc-binding motif (D⁹³DGH<u>DH</u>⁹⁸) in its C-terminus. Our data indicate that PacL proteins co-localize in shared foci, and can complement each other functionally, at least partially and depending on the surrounding metal microenvironment.

From an evolutionary viewpoint, it is interesting that PacL proteins show homology with the C-terminal end of the mycobacterial chaperone GroEL1. GroEL1 contains a C-terminal histidine-rich region, which is a general feature of GroEL1 in actinobacteria. Recent studies showed that this region is involved in metal binding, which is required for GroEL1 function[66,67], and it is possible that GroEL1 and PacL proteins evolved from a common ancestor to fulfill distinct functions.

Finally, although DUF1490-containing proteins are present exclusively in actinobacteria, including mycobacteria, they are related to DUF6110-containing proteins (http://pfam.xfam.org/family/DUF6110), which exhibit similar predicted structural features but lack a predicted metal-binding motif (Supplementary Fig. 12a). DUF6110 are mainly present in firmicutes; yet they are also present in other bacterial phyla, including Actinobacteria and Proteobacteria, as well as in Archaea (Supplementary Data 1 and Supplementary Fig. 12b). In silico analyses revealed that DUF6110 proteins are typically encoded together with transporters of the poorly characterized $P_{1B6}$ family[16], suggesting they might be involved in membrane platform assembly of these P-ATPases (Supplementary Data 1 and Supplementary Fig. 12b). CtpC exhibits features of $P_{1B6}$-ATPases, such as the presence of a highly conserved HN motif in $TM_6$ (Supplementary Fig. 12c, d); however, it also contains a conserved CPC motif in $TM_4$, a typical feature of $P_{1B1}$- and $P_{1B2}$-ATPases (Supplementary Fig. 12d). Therefore, CtpC could be considered a "$P_{1B6}$-like-ATPase". Most DUF1490-containing proteins are encoded with such $P_{1B6}$-like-ATPases (Supplementary Data 1 and Supplementary Fig. 12b). Beyond *M. tuberculosis*, it remains to be evaluated whether features of PacL proteins are conserved among DUF1490- and DUF6110-containing proteins, which might provide insight into the regulation and function of $P_{1B6}$- and $P_{1B6}$-like-ATPases.

Given the high significance of metal efflux in environmental and pathogenic microbes, understanding the biology of bacterial membrane efflux platforms will undoubtedly open up promising avenues for future research in the development of novel antimicrobials and to better understand the physiology of environmental prokaryotes.

## Methods

### Bacterial strains and culture conditions

*M. smegmatis* mc²155 (ATCC 700084), *M. tuberculosis* H37Rv (ATCC 27294) and mutants or recombinants derived from these strains were grown at 37 °C in 7H9 medium (Difco) supplemented with 10% albumin-dextrose-catalase (ADC, Difco) and 0.05% Tween-80 (Sigma-Aldrich), or on complete 7H11 solid medium (Difco) supplemented

with 10% oleic acid-albumin-dextrose-catalase (OADC, Difco). Of note, 7H9 medium contains 6 μM ZnSO$_4$. When required, streptomycin (25 μg.ml$^{-1}$), hygromycin (50 μg.ml$^{-1}$) or zeocin (25 μg.ml$^{-1}$) were added to the culture media. *Escherichia coli* strains Stellar™ (Takara bio) and Rosetta™ 2 (Novagen) were grown at 37 ˚C in LBroth (Difco) or on L-agar plates (Difco) supplemented with streptomycin (25 μg.ml$^{-1}$) or ampicillin (100 μg.ml$^{-1}$) when required.

### Construction of *M. tuberculosis* and *M. smegmatis* mutants
Mutant strains of *M. tuberculosis* H37Rv and *M. smegmatis* mc²155 were constructed by allelic exchange using the recombineering technology[68], as we described previously[55]. Two ~0.5-kb DNA fragments flanking the genes of interest were amplified by PCR using PrimeSTAR GXL DNA polymerase (Takara bio), *M. tuberculosis* H37Rv or *M. smegmatis* mc²155 genomic DNA, and appropriate primer pairs (Supplementary Table 1). A three-fragment PCR fused these two fragments to a zeocin-resistance cassette, either intact, to generate deletion marked by a zeocin-resistance gene, or flanked by variants of the *M. tuberculosis* *dif* site, to generate unmarked deletions[55]. The allelic exchange substrate (AES) was recovered by agarose gel purification. The recipient strains for recombineering were derivatives from *M. tuberculosis* H37Rv or *M. smegmatis* mc²155 carrying the plasmid pJV53H, a hygromycin-resistant pJV53-derived plasmid expressing recombineering enzymes[68]. These strains were grown in complete 7H9 medium supplemented with hygromycin until mid-log phase and expression of recombineering enzymes was induced by 0.2% acetamide (Sigma-Aldrich). After induction, electrotransformation was performed with 100 ng of the linear AES for allelic exchange. After 4 h (*M. smegmatis*) or 48 h incubation at 37 °C (*M. tuberculosis*), mycobacteria were plated onto agar plates supplemented with zeocin. Zeocin-resistant clones were plated on the same medium, and single colonies were grown in complete 7H9 without antibiotic and verified to carry the expected allele replacement by PCR on chromosomal DNA, using external primers. Spontaneous loss of the *dif* sites-flanked-zeocin-resistance cassette by XerCD-dependent recombination and of the pJV53H plasmid were obtained by serial rounds of culture without antibiotics and phenotypic tests for zeocin and hygromycin sensitivity.

### *M. smegmatis* and *M. tuberculosis* expression vectors
The *pacL1* gene (Rv3269) or the *pacL1-ctpC* operon (Rv3269-Rv3270) preceded by their native promoter were amplified by PCR using PrimeSTAR GXL DNA polymerase, *M. tuberculosis* H37Rv genomic DNA as template, and primer pairs clo-B2-3269-Am/clo-B3-3269-Av or clo-B2-3269-Am/clo-B3-ctpC-Av, respectively (Supplementary Tables 2 and 3). Plasmids pGMCS-pacL1 and pGMCS-pacL1-ctpC were constructed by multisite gateway recombination[69], using plasmid pDE43-MCS as destination vector. These plasmids are integrative vectors (insertion at the *attL5* mycobacteriophage insertion site in the *glyV* tRNA gene) and express PacL1 or PacL1 and CtpC under the control of their zinc-inducible native promoter[12].

The derivative plasmids with deletion, substitution or tagged version were constructed by in-fusion cloning. PCR fragments were amplified using appropriate primer pairs (Supplementary Tables 2 and 3) and Phusion High-Fidelity PCR Master Mix with GC Buffer (Thermo Scientific). Linear fragments were purified on agarose gels, and circularized using the In-Fusion HD Cloning Kit (Takara), following the manufacturer's instructions. Plasmids transformed into Stellar recipient cells were verified by sequencing and introduced by electroporation into *M. tuberculosis* H37Rv or *M. smegmatis* mc²155 strain.

### Mycobacterial zinc sensitivity test
*M. tuberculosis* cultures were inoculated at OD$_{600}$ 0.01 in glass tubes in complete 7H9 medium containing different ZnSO$_4$ concentrations. Cultures were incubated without agitation at 37 °C and growth was monitored by measuring turbidity (expressed in McFarland units) over time using a Densimat apparatus (BioMerieux).

*M. smegmatis* cultures were inoculated at OD$_{600}$ 0.02 in complete 7H9 medium containing different ZnSO$_4$ concentrations. Cultures were incubated for 24 h at 37 °C with agitation and the OD$_{600}$ was measured. Bacterial growth was expressed as percentage relative to the condition without zinc.

*M. tuberculosis* cultures were inoculated at OD$_{600}$ 0.01 in complete 7H9 medium containing 500 μM ZnSO$_4$ and incubated at 37 °C. At the indicated time points sample were collected, diluted and plated onto 7H11 medium, and CFU were scored after 21 days.

### RNA extraction and RT-qPCR
Total RNA was extracted from cultures grown to logarithmic phase (OD$_{600}$ between 0.6 and 0.8) using the RNeasy kit (Qiagen) following the manufacturer's instructions with minor modifications[70]. RNA samples were treated for 30 min with 2U of Turbo DNase (Turbo DNA free kit, Ambion). The amount and purity of RNA were quantified using a NanoDrop ND-1000 apparatus (ThermoFischer Scientific) by measuring absorbance at 260/280 nm. Double-stranded cDNA was reverse-transcribed using the superscript III Reverse Trancriptase kit (Invitrogen), according to the manufacturer's protocol. For real-time qPCR, specific primers were designed and PCR reactions were performed using SYBR Green Premix Ex Taq (Takara), according to the manufacturer's protocol and the primers listed in Supplementary Table 4. All real-time qPCR reactions were carried out using a 7500 Real-Time PCR System and data were analyzed using the 7500 Software version 2.3 (Applied Biosystems). The endogenous control gene was *rpoB* for *M. tuberculosis* and *sigA* for *M. smegmatis*.

### Membrane preparation and Western blotting experiments
Mycobacterial cultures for WB were performed in 7H9 medium with Tween-80 but without ADC, and subcellular fractions were obtained as previously described[71]. 50 mL cultures were centrifuged for 10 min at 3000 × *g* at 4 °C, and washed twice with PBS containing protease inhibitor (cOmplete™, EDTA-free protease inhibitor cocktail, Roche). Bacteria were lysed by the addition of 0.1 μm-diameter glass beads followed by four 60-s cycles pulses at full speed in a bead-beater device. The lysate was subject to 1 h of centrifugation at 27,000 × *g* at 4 °C. The supernatant was subject to ultra-centrifugation for 1 h at 100,000 × *g* to separate the soluble fraction and the membrane-enriched pellet fraction.

Proteins were separated in polyacrylamide gels (NuPAGE 4 to 12%, Bis-Tris, 1.0 mm, Mini Protein Gel, 12-well, Invitrogen), and transferred onto nitrocellulose membranes (Trans-Blot Turbo RTA Midi Nitrocellulose Transfer Kit, Bio-Rad).

To perform Blue-native (BN) polyacrylamide gel electrophoresis (PAGE), mycobacterial cultures were prepared under the same condition as for SDS-PAGE; after bead beater treatment, the lysate was subject to 1 h of centrifugation at 3000 × *g* at 4 °C. The supernatant was subject to ultra-centrifugation for 30 min at 100,000 × *g* at 4 °C and membrane proteins were solubilized for 30 min at 4 °C with 0.25% n-Dodecyl β-D-maltoside (DDM, Sigma). The insoluble fraction was removed by centrifugation at 18,000 × *g* for 30 min at 4 °C and solubilized proteins were separated on a BN-PAGE (3–12%, Bis-Tris, 1.0 mm, Mini Protein Gels, Invitrogen) and transferred onto PVDF membranes (Invitrolon 0.45 μm, Invitrogen).

Membranes were blocked in 5% BSA in TBS Tween-20 0.05% and incubated overnight with corresponding primary antibodies at 4 °C. HRP-coupled antibodies were used for the detection of the Flag and HIS$_6$ tags using anti-Flag (Monoclonal Anti-Flag clone M2-Peroxidase (HRP), Sigma, 1/30,000) and anti- HIS$_6$ (6x-His Tag Monoclonal Antibody (clone HIS.H8), HRP, Thermo Fisher, 1/5,000). For the detection of mTurquoise and mVenus fluorescent proteins, anti-eGFP was used (eGFP Monoclonal Antibody, clone F56-6A1.2.3, Thermo Fisher, 1/

1,000), associated with an anti-mouse secondary antibody (Goat antiMouse IgG (H + L) Secondary Antibody, HRP, Thermo Fisher, 1/10,000). When needed, membranes were stripped for 10 min in 200 mM Glycine, 3.5 mM SDS, 1% Tween-20 at pH 2.2, and a second antibody was used. Signals were detected by autoradiography using HRP reaction (WesternBright Quantum HRP Substrate for CCD, Diagomics) on a ChemiDoc Touch Imaging System (BioRad). Unprocessed and uncropped WB are provided in Supplementary Fig. 13.

### Zinc quantification in mycobacteria

*M. smegmatis* was cultured in 7H9 medium containing Tween-80 but without ADC. When $OD_{600}$ reached 0.5, $ZnSO_4$ (100 μM) was added or not (control) to the cultures. After 1 h incubation at 37 °C, bacteria were washed 2 times in 7H9 medium. FluoZin-3 (FluoZin™−3, AM, cell permeant, Thermo Fisher) was added at 1 mM, and cultures were incubated for 1 h at 37 °C.

Samples were washed, recovered in PBS and observed by fluorescence-activated cell sorting (FACS); data were analysis using FlowJo (v10) software.

### FACS analysis of fluorescent reporters

*M. smegmatis* mc²155 cultures were inoculated in complete 7H9 medium at $OD_{600}$ 0.03 with different concentrations of metals ($ZnSO_4$, $MnSO_4$, $CuSO_4$). After incubation overnight at 37 °C bacteria were collected and observed by FACS; data were analysis using FlowJo (v10) software. The mCherry signal was used as basal level of expression. FACS was performed using an LSR Fortessa (BD Biosciences) flow cytometer. Gating strategy is displayed in Supplementary Fig. 14.

### Fluorescence microscopy

Mycobacterial cultures expressing fluorescent plasmids were applied as 0.6 μL drops to the surface of a layer of 1% agarose in growth medium, as we described previously[72]. The cells were imaged at 30 °C using an Eclipse TI-E/B wide field epifluorescence microscope with a phase contrast objective (CFI Plan APO LBDA 100X oil NA1.45) and a Semrock filter YFP (Ex: 500BP24; DM: 520; Em: 542BP27), CFP (Ex: 438BP24; DM: 458; Em: 483BP32) or FITC (Ex: 482BP35; DM: 506; Em: 536BP40). Images were taken using an Andor Neo SCC-02124 camera with illumination at 80% from a SpectraX source Led (Lumencor) and exposure times of 0.5−1 s. Nis-Elements AR software (Nikon) and ImageJ (v 1.53f51) were used for image capture and editing.

### Purification of soluble PacL1 and variants

TMHMM (TMHMM Server v. 2.0) was used as a membrane protein topology prediction method based on a hidden Markov model (HMM), to delimit the soluble domain of PacL1 (SolPacL1), from Ala[30] to His[93]. The corresponding DNA fragment was amplified using the primers SolPacL1Fw and SolPacL1Rv (Supplementary Table 3), using *M. tuberculosis* genomic DNA as template. The PCR product was subcloned into the pProEx-Htb expression vector (Invitrogen) at the NcoI/HinDIII sites, downstream the [HIS_6/spacer/TEV protease site] coding sequence.

Subcloning of the sequence coding for SolPacL1, deleted from the last three amino acids (SolPacL1Δ3), was made following the same strategy using the primers SolPacL1Δ3-Fw and SolPacL1Δ3-Rv (Supplementary Table 3). The correct insertion of SolPacL1- and SolPacL1Δ3-coding sequences in the pProEx-HTB vector was verified by sequencing.

The production of SolPacL1 and SolPacL1Δ3 was performed in the *E. coli* strain Rosetta2. After an overnight preculture in LB supplemented with 50 μg/mL ampicillin, 1% (w/v) glucose, 1 mM $MgCl_2$, 7.5 mM $CaCl_2$ at 37 °C, a 1 L culture was started at an $OD_{600}$ of 0.1, in the same medium, at 37 °C. When the $OD_{600}$ reached 0.6−0.7, the production of the protein of interest was induced by addition of

1 mM IPTG. After 2 h, cells were harvested by centrifugation (5 min, 16,800 × *g*, 4 °C). Cells were resuspended in the lysis buffer (50 mM $NaH_2PO_4$, pH 8.0, 300 mM NaCl, 50 mM Imidazole, 10 % (v/v) glycerol, 1 mM Phenylmethanesulfonyl fluoride (PMSF), EDTA-free protease inhibitor (1 tablet / 200 mL buffer, Roche) and lysed by sonication. After centrifugation (73,000 × *g*, 30 min, 4 °C), the supernatant was loaded onto a Ni-NTA column equilibrated in 50 mM $NaH_2PO_4$, pH 8.0, 300 mM NaCl, 50 mM imidazole (equilibration buffer). The column was washed with 200 mL of the equilibration buffer supplemented with 1 mM PMSF and EDTA-free protease inhibitor (Roche). SolPacL1 and SolPacL1Δ3 were then eluted with 25 mL of the equilibration buffer containing 1 mM PMSF and 250 mM imidazole.

The cleavage of the HIS_6 tag of SolPacL1 and SolPacL1Δ3 was initiated by the addition of 8 μg/mL His-tagged TEV protease, in the elution buffer supplemented with 100 μM EDTA, 1 mM DTT, 100 μM PMSF. To remove imidazole, the reaction was performed in a dialysis bag (overnight, 4 °C) against 2 L of a buffer containing 50 mM $NaH_2PO_4$, pH 8.0, 150 mM NaCl, 100 μM EDTA, 1 mM DTT, 100 μM PMSF. To remove EDTA and DTT, a second dialysis step (4 h, 4 °C) was performed against 2 L of a buffer containing 50 mM $NaH_2PO_4$, pH 8.0, 300 mM NaCl, 100 μM PMSF. After cleavage by the TEV protease, the sample, supplemented with 50 mM imidazole, was loaded onto a Ni-NTA column. SolPacL1 and SolPacL1Δ3 rid of the HIS_6 tag were recovered in the flow-through. Proteins were concentrated using Vivaspin-PES (GE HealthCare, cut-off 5 kDa), loaded onto a Superdex 75 column (GE HealthCare) and eluted in 50 mM MOPS, pH 7.0, 200 mM NaCl. The proteins having no aromatic residues, chromatography was followed at 214 nm. Fractions containing the protein of interest were identified by SDS-PAGE (4−20% precast polyacrylamide gel, BioRad), pooled and concentrated using Vivaspin-PES (cutoff 5 kDa). Routinely, 1 mg of pure protein was obtained per liter of culture.

### Production of PacL1-derived peptides

N-α-Fmoc-protected amino acids, 2-chlorotrityl chloride resin and HCTU coupling reagent were obtained from Novabiochem or Iris Biotech. Other reagents for peptide synthesis and solvents were purchased from Sigma-Aldrich.

Analytical HPLC separations were performed on an Agilent Infinity 1260 system using a Waters Xbridge BEH C18 (2.5 μm, 75 mm × 4.6 mm) column at a flow rate of 1 mL/min. Preparative HPLC separations were performed on a VWR LaPrepΣ system using a Waters XBridge Peptide BEH130 C18 (5 μm, 150 mm × 19 mm column at flow rates of mL/min. Mobile phase consisted in a gradient of solvent A (0.1% TFA in H2O) and B (0.1% TFA in MeCN/H2O 9:1). For analytical separations, Method A consisted in 5% B during 2 min followed by a 5−70% B linear gradient in 13 min at 1 mL/min. Eluate was monitored by electronic absorption at 214 and 280 nm. LRMS (ESI+) analyses were performed on a Thermo LXQ spectrometer.

For the synthesis of Trp-MBM (Ac-WDLHDHDH-OH), a solution of Fmoc-L-His(Trt)-OH (93 mg, 150 μmol) and DIEA (150 μL) in DCM (4 mL) was added to 2-chlorotrityl resin (100–200 mesh, 600 mg, 1.4 mmol/g) and stirred for 1.5 h. The resin was washed with DCM and DMF, capped with a DCM/methanol/DIEA (7:2:1 v/v/v, 5 mL) mixture for 20 min then washed with DMF and DCM. Peptide elongation was performed using standard solid phase peptide synthesis procedures using Fmoc/tBu chemistry on an automated peptide synthesizer (CEM Liberty1 Microwave Peptide Synthesizer). Double couplings (30 min) were performed using 3-fold molar excess of Fmoc-L-amino acid, 2.7-fold molar excess of HCTU and 6-fold molar excess of DIEA at room temperature. A capping step was performed after each coupling with Ac_2O/DIEA in DMF (5 min). Fmoc removal was performed using 20% piperidine in DMF (2 × 10 min). After the coupling of the last amino acid, the Fmoc protecting group was

removed manually using 20% piperidine in DMF (3 × 5 min) and the N-terminus was acetylated using a DMF/pyridine/Ac₂O (7:2:1 v/v/v, 5 mL) mixture (5 min). The resin was washed with DMF and DCM then dried. Removal of acid-labile side chain protecting groups and cleavage of the peptidyl resin was performed with TFA/H₂O/TIS (9.4:0.3:0.3 v/v/v, 10 mL) for 2 h. The acidic solution was collected and concentrated under reduced pressure and cold diethyl ether was added to precipitate the peptide. After HPLC purification and lyophilization, Trp-MBM was obtained as a pure white powder (69 mg). HPLC (anal.): $t_R$ = 4.9 min; LRMS (ESI+): monoisotopic m/z = 1116.3 (+), 558.7 (2+)/calculated monoisotopic $m/z$ = 1116.45 [M + H]+, 558.73 [M + 2H]²+ for M = $C_{49}H_{61}N_{15}O_{16}$.

### Dialysis at equilibrium

The binding stoichiometry of the indicated metals was determined by dialysis at equilibrium using dialysis cassettes from Thermo Fisher (Slide-A-Lyzer™ 2 K MWCO). In the protein chamber, the concentration of apo-solPacL1 (or apo-solPacL1Δ3) was set at 50 μM in 500 μL of 50 mM MOPS 50 mM NaCl pH 7. For the experiment, performed at 4 °C for 21 h, the dialysis cassette was immersed in 140 mL of the same buffer supplemented with 50 μM of a given metal. The amount of metal bound to the protein was determined using 4-(2-pyridylazo) resorcinol (PAR), following the absorbance at 495 nm, due to the formation of the metal–PAR2 complex.

### Size Exclusion Chomatography-Multi-Angle Light Scattering (SEC-MALS)

SEC was run on a SuperdexTM 200 Increase 10/300GL column (GE HealthCare) at 0.5 mL/min in 25 mM Tris-HCl pH8.8, 500 mM NaCl. MALS analysis was performed on the coupled instruments DAWN HELEOS-II and Optilab Rex (WYATT).

### Circular dichroism (CD)

CD experiments were performed on 30 μM of the apo forms of solPacL1 and SolPacL1Δ3 in 10 mM MOPS pH 7.2 using a J-815 CD spectrometer (JASCO) at a scan speed of 100 nm/min, from 250 to 190 nm. CD experiments on the Trp-MBM were performed on 50 μM of peptide in water adjusted to pH 7 using a Chirascan CD spectrometer (Applied photophysics). Refractive index increment dn/dc was set at 0.185 mL/g.

### Isothermal Titration Calorimetry (ITC)

ITC experiments were performed at 25 °C with a PEAQ-ITC isothermal titration microcalorimeter (Malvern Instruments, Malvern, UK). The sample (200 μL) containing 150 μM of purified solPacL1 or of Trp-MBM in 50 mM MOPS pH7, 200 mM NaCl was first placed in the microcalorimeter cell. Titration was performed by successive additions every 2 min of 2 μL of 1.5 mM Zn[CH₃COO]₂. The experimental data were fit to a theoretical titration curve using the supplied MicroCal PEAQ-ITC analysis software, with ΔH (enthalpy change), Kd (dissociation constant) and N (number of binding sites per monomer) as adjustable parameters. Free energy change (ΔG) as well as entropy contributions (TΔS) were derived from the equation ΔG = ΔH − TΔS = −RT ln(Ka) (where T is the absolute temperature, $R$ = 8.314 J.mol⁻¹.K⁻¹ and Ka = 1/Kd).

### Peptide fluorescence analysis

Tryptophan intrinsic fluorescence of Trp-MBM was measured using a Cary Eclipse spectrophotometer (VARIAN), in 2.4 mL of 10 mM HEPES pH7.5 containing 20 μM of the peptide. Excited wavelength: 280 nm; Maximum emission wavelength 357 nm.

### Nuclear Magnetic Resonance (NMR)

NMR spectra were performed on Bruker AvanceIII HD spectrometer operating at a ¹H Larmor frequency of 700.13 MHz at 7 °C. The NMR experiments were performed on PacL1 samples at 500 μM in 20 mM MES-d₁₃ pH 6.0 and pH 7.0, 100 mM NaCl with 106 μM of sodium 3-(trimethylsilyl)propane-1-sulfonate (DSS) as internal reference. PacL1 samples with zinc were prepared similarly but with 1 mM of zinc acetate (2 Eq. of Zn). NMR experiments were recorded with an excitation sculpting scheme to suppress the water signal[73] using a selective pulse (1 ms length and shape: sinc1.1000). All pulses were calibrated with respect to the 90° hard pulse calibrated at 8.25 μs. 1D ¹H spectra were recorded with 16 scans, an acquisition time of 1.7 s (spectral width 16 ppm) and a relaxation delay between scans of 2 s. Assignment was achieved using TOCSY and NOESY experiments with mixing times of 80 ms (DIPSI2 with a RF field of 10 kHz) and 120 ms respectively[74]. Two dimensional spectra were acquired with 8 K and 1 K complex data points in direct and indirect dimensions respectively (spectral width 12 ppm). Experiments were processed with Topspin (Bruker Biospin) and the peaks were assigned manually using the program CARA[75]. Almost 70% of the residues were assigned unambiguously. An alanine-rich moiety at the N-terminus (i.e., A³⁰AAKAP[35]) and twelve residues from R52 to L63 (i.e., R⁵²KAEEAAESARL[63]) could not be assigned due to severe peak overlap. Chemical shift index values were determined from Hα chemical shift as previously described[43]. The chemical shift perturbation of amide protons (Hₙ CSP) induced by zinc binding was measured as the absolute value of the difference in chemical shifts with and without zinc, |Hₙ ₍₊₎ Zinc − Hₙ ₍₋₎ Zin|. Chemical shifts were referenced to DSS.

### Statistical analyses

Data were analyzed using the one-way ANOVA, or two-tailed $t$-test after validating data normality distribution using the Shapiro–Wilk and/or Kolmogorov–Smirnov normality tests, as indicated in the figure legends. Data were analyzed using GraphPad Prism 9.3.1 (GraphPad software). No statistical method was used to predetermine the sample size. Throughout the study, the sample size was determined based on our preliminary studies and on the criteria in the field. Biological replicates ($n$) and the numbers of the independent experiments are indicated in each figure legend.

### Biological materials

All unique materials are readily available from the corresponding authors on request.

### Reporting summary

Further information on research design is available in the Nature Research Reporting Summary linked to this article.

## Data availability

Raw data are provided as a Source Data file. Raw images and NMR spectra are accessible on Mendeley, https://data.mendeley.com/datasets/c7zdzrbs68/1. Source data are provided with this paper.

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

## Acknowledgements

We thank Yannick Poquet (IPBS, Toulouse) and Julien Vaubourgeix (Imperial College, London, UK) for helpful discussion and critical reading of the manuscript. We thank Life Science Editors for editorial assistance. We thank Emmanuelle Näser and Penelope Viana (Génotoul TRI-IPBS platform, Toulouse), and Denis Hudrisier for help with flow cytometry, Brice Ronsin (Genotoul TRI-CBI, Toulouse) and Serge Mazères (Genotoul TRI-IPBS platform, Toulouse) for help with microscopy, Pascal Ramos (Genotoul PICT-IPBS platform, Toulouse) for technical support with NMR, and Anne Imberty (Centre de Recherches sur les Macromolécules Végétales (UPR5301) CERMAV-CNRS, Grenoble) for help with ITC. We thank Dirk Schnappinger (Weill Cornell Medicine, New York, NY) for kindly providing plasmids. This work received funding support from the Centre National de la Recherche Scientifique (CNRS), the University of Toulouse-University Toulouse III Paul Sabatier, the Fondation pour la Recherche Médicale (FRM, fellowship to Y.M.B and grant n° DEQ20160334902 to O.N.), the JPI-AMR program (grant n° JPIAMR_2018_P010 to O.N.), the Fondation Bettencourt Schueller (Prix Coups d'élan to O.N. and Expore-TB grant top O.N.), the Agence Nationale de la Recherche (ANR, grant n° ANR-14-CE14-0024 to O.N. and P.C., and grant n° ANR-21-CE11-0031 to O.N.) and MSDAVENIR (grant FIGHT-TB to O.N.). We acknowledge support from the Labex ARCANE, CBH-EUR-GS, ANR-17-EURE-0003. Bruker Avance III HD 700 MHz NMR spectrometer of the Integrated Screening Platform of Toulouse (Genotoul PICY-IPBS platform) was funded by CNRS, the University of Toulouse-University Toulouse III Paul Sabatier, IBiSA, European Structural Funds, and the Occitanie Region.

## Author contributions

Conceptualization, Y.M.B., P.C., C.G., O.N. Investigation, Y.M.B., M.F., X.M., Y.R., R.M., P.D., O.Sé., O. Sa., J.R., M.W., P.C., C.G., O.N. Writing, O.N. with contribution from Y.M.B., O.Sa., P.C., C.G. Funding acquisition, J.Y.B., P.C., O.N. Supervision, J.Y.B., P.C., C.G., O.N.

## Competing interests

The authors declare no competing interests.
