## [Peer Review File · Nature Communications]

Reviewers' Comments:

Reviewer #1:

Remarks to the Author:

In the report, Mycobacterial resistance to zinc poisoning requires assembly of P-ATPase-containing membrane metal efflux platforms, Yves-Marie Boudehen et al., describes a new type of chaperone protein to cluster membrane-associated metal efflux pumps that contribute to *M. tuberculosis* infection. This referee acknowledges the amount of work and the difficulty to work with the bacterium *M. tuberculosis* in laboratory conditions. There are however several issues to address:

The concept of "membrane microdomain" or "membrane platform" it is not entirely clear to this referee and I think it would be important to clarify this matter. How are these concepts different from classical membrane-associated protein complexes? As far as I understand, the concept of membrane microdomain requires the coalescence of membrane lipids and proteins whereas the term "membrane platform" is rather unclear to me. It is also difficult for me to draw a line between a platform and a typical membrane-bound protein complex. The text uses indistinctly these terms and it is at times rather confusing. I did not see experiments involving membrane lipids so the authors should explain very carefully the use of the term "membrane microdomain". Alternatively, they can perform additional experiments to show a connection between Pacl1 and certain membrane lipids, to fully demonstrate that they coalesce into micro/nanodomains.

I missed infection experiments to demonstrate that mutations in Pacl1 phenocopy the infection defect that the authors have reported in mutants defective in the metal efflux pumps. One would hypothesize that, as Pacl1 is required for the stability of the metal pumps, a mutant in Pacl1 is unable to cause an infection, as mutants in the metal pumps are infection defective mutants. I find these infection experiments important, to demonstrate there is a connection between Pacl1 and the efflux pumps but also to demonstrate the physiological importance of Pacl1 and the relevance of the formation of membrane microdomains/platforms in mycobacterial virulence.

Minor comments

Are Pacl2 and Pacl3 structurally similar to Pacl1? It seems this is not the case. Could the authors comment in the discussion section on the possible roles of these proteins?

The authors should include an additional control to show that the constructs to fluorescent proteins are functional and localize correctly in the bacterial membrane.

Fig 5C. The cells express a large amount of Flag-tagged Pacl1. This may generate artefactual interactions.

Reviewer #2:

Remarks to the Author:

In this manuscript, Boudehen et al. present an unusually comprehensive investigation into the function of a member of an enigmatic class of small single-pass TM proteins, denoted Pacl1 (DUF1490), that is transcriptionally induced by high Zn(II) alongside the P1B-type ATPase metal effluxer, CtpC, in the human pathogen *Mycobacterium tuberculosis*. Resistance to host-mediated zinc toxicity in *M. tuberculosis* is particularly important in the phagocytic compartment in infected cells, and this is strongly worthy of detailed study. The findings reported here are novel, surprising (notably, a distant evolutionary connection to *Mtb* GroEL and DUF6110 proteins), and of broad significance, which should catalyze a host of follow-up studies in the bacterial metal homeostasis and resistance fields.

The work presented here sheds considerable new light on the structure and function of the Pacl1 and their related orthologs in the *Mtb*, Pacl2 and Pacl3. The authors present compelling evidence in support of the hypothesis that CtpC is strictly dependent on Pacl1 for functional localization to the mycobacterial membrane and metal efflux, in what could be considered dynamic microdomains. The authors tease apart and functionally segregate the domains of this ~93-residue

PacL1 (with clear parallels to PacL2 and PacL3), and provide compelling evidence in support of an E/A-rich domain that is required for membrane location by PacL1, and a C-terminal Zn(II) coordination site that becomes functionally important only at high Zn(II) stress (Fig. 4D) (a striking finding). NMR studies suggest a rapidly interconverting coordination environment for the metal, which is not surprising given this short, linear metal-binding motif. They further show that PacL orthologs are at least partially redundant (but not entirely so). Finally, coincident localization of PacL1 and CtpC to the membrane occurs independent of the metal-binding site and the mycobacterial flotillin FloT, with PacL1 essentially functionally substituting for FloT.

I have only a few comments the authors may wish to consider:

1) p. 5, line 147: Please check the units on the radius of gyration; 3 nm is likely correct, consistent with an extended polypeptide chain.

2) p. 7, line 232: Do Msm $\Delta 2$ cells expressing CtpC with PacL1 lacking E/A-rich repeats accumulate intracellular Zn as evidenced by FluoZn staining? This control appears to be missing.

3) Fig. 4B, ITC: Please correct these data for the addition of Zn into buffer, and re-plot and re-fit the data. The net heat should go to 0 when the peptide is saturated with metal. Or is there another binding event here? Please indicate the fitted parameters and stand errors for this experiment (average of all experiments) in the figure legend.

4) p. 6, line 217: Please provide a reference after the word "metallochaperone"

5) Fig. 6D, line 1045: I find it puzzling that the authors did not do a parallel series of experiments with the PacL3. Why not? One might anticipate full Cu-dependent complementation and rescue from Zn toxicity in the Δ pacL1 strain.

6) What's left for future work is a mechanistic unpacking what appears to be co-localization phenomenon of the PacL1 and CtpC. In my mind the authors provide no strong support for or against a metallochaperone model (direct metal transfer by ligand exchange), and the absence of detection of even a transient physical interaction between the two proteins, would seem to argue for a local metal concentration or sequestration effect. Perhaps the discussion section can be revised to better reflect these ideas.

7) Can the authors comment on the mode of Zn-inducible transcription of the pacL1-ctpC?

Reviewer #3:

Remarks to the Author:

Manuscript: NCOMMS-21-35884-T

Reviewer comments

Key results:

This work addresses a relevant topic in mycobacteria: the molecular mechanisms activated by M. tuberculosis to overcome toxic levels of heavy metals inside macrophages. The possible role of some P-type ATPases in heavy metal detoxification processes and ion specificity to exert mycobacterial resistance to metal poisoning is known. However, there is lack of information about the correlation of P-type ATPases with others cell components, such as metallochaperones, involved in the ion homeostasis of the plasma membrane.

The results presented in this work are valuable because, for the first time, other membrane components are associated with the P-type ATPase activity (in this case mediated by CtpC) in the mycobacterial cell membrane is evidenced. The existence of this putative new family of proteins (PacLs) is postulated, being part of the same operon of different mycobacterial P-type ATPases: CtpC, CtpG or CtpG.

Specifically, this work suggests that the Pacl1 interaction with CtpC is necessary to prevent Zn²⁺ accumulation inside mycobacteria. In addition, structural elements in Pacl1 necessary for the CtpC activation and function to exert mycobacterial resistance to Zn²⁺ poisoning are even described. The presence of flotillin-independent FMMs that underlies mycobacterial resistance to metal poisoning is proposed, a novel mechanism at date unknown for mycobacteria.

Validity:

This manuscript has a correct execution; therefore, I consider there is not reason to prohibit its publication.

Originality:

According to the literature I have checked to date, the conception of this work is original, together with its execution, results, discussion and conclusions.

Significance:

Since much information is still lacking of the mycobacterial P-type ATPases functioning, I sincerely consider that the results presented in this work are interesting and valuable. Since MMFs have already been described in bacteria, there is no urgency in the publication of this manuscript. In addition, it seems that the mycobacterial resistance to Zn²⁺ poisoning and this cation translocation across the cell membrane is not exclusive to P-type ATPases.

Abstract:

The text of the abstract is clear, however. The main purpose of the manuscript is to describe and report a new type of protein, which is not fully included in the abstract.

Introduction:

Although the main objective of the study is apparently to describe the Pacl1 proteins in mycobacteria, the same study links those closely to the function of the mycobacterial cell membrane transporter CtpC. In this sense, it is strange that there is no reference to this topic in the introduction (Padilla-Benavides et. al, JBC, Vol. 288, 16: 11334 –11347, 2013) where in a elegant way, it is demonstrated that Mn²⁺ or Ni²⁺ (to a lesser extent) can be CtpC substrates in mycobacterial cells. It is necessary to clarify which of these two cations is the main CtpC substrate in vitro and in vivo, in *M. tuberculosis* and *M. smegmatis*, according to the known information; additionally, to clarify the possible role of CtpC in the mycobacterial resistance to toxic Zn²⁺ concentrations (Cell Host & Microbe 10, 248–259, September 15, 2011). To begin with, it is very relevant evidence that must be taken as part of the approach of this work.

Data & methodology:

I consider the data reported and the methodology presented in this manuscript are sufficiently detailed and clear to allow the reproduction of the results.

The strategy proposed to solve the different research questions is adequate; the experiments are sound, and usually the data obtained are reliable. Here a description of the most relevant experiments according to the reviewer's point of view.

The western blot and kinetic of growth experiments showed in figure 2 clearly demonstrate that the Pacl1 and CtpC expression in *M. tuberculosis* and *M. smegmatis* is mediated by Zn²⁺. In addition, both, Pacl1 and CtpC are necessary for Zn²⁺ efflux under toxic concentrations of this cation, being essential the TM domain of Pacl1. Western blotting, Flow cytometry and Eclipse TI-E/B wide field epifluorescence experiments in Figure 2 and 3, also demonstrate the colocalization of Pacl1 and CtpC in the plasma membranes of *M. tuberculosis* and *M. smegmatis*.

Regarding to the binding stoichiometry of Pacl1/Zn²⁺, the equilibrium dialysis and ITC experiments partially demonstrate that Pacl1 binds Zn²⁺ (not Mn²⁺) in a 1:1 molar ratio, and mediated by Hys residues at the MBM of Pacl1. This finding is also corroborated by NMR spectroscopy. Therefore, it is clear that Pacl1 MBM is required for mycobacterial resistance to zinc

at higher concentrations.

Preliminary evidence that Pacl1 could be a flotillin-like protein is observed in immunofluorescence, immunoblotting and qRT-PCR experiments showed in Figures S5 and S7. Specifically, results suggest that the Glu/Ala repeats in Pacl1 cytoplasmic region is required for abundant CtpC levels in the plasma membrane, and for the proper function of Pacl1 triggering resistance to Zn²⁺ excess. The immunofluorescence experiments also demonstrate that Pacl2 and Pacl3 colocalize in the cell membrane and are associated to bacterial resistance to metal poisoning.

Detailing the Pacl1 portion involved in the Zn²⁺ resistance, the immunofluorescent label in Figure 6 indicates that the Pacl1 MBM is required for resistance to high concentration of Zn²⁺. In addition, that CtpC is necessary (I would not say essential) for Zn²⁺ resistance.

Use of statistics and treatment of uncertainties:

Although the number of processed samples in some experiments allows a clear statistical significance (e.g. Fig 1E, 6E), by contrast, the error from bars is not quantified in other cases (e.g. Fig 1B, 1G, 3I, 4D, 5A, 5D, 5E, 6D, etc). On the other hand, the respective standard deviation for values obtained by means, always should be reported. For this reason, the number of dates obtained from processed samples have to be increased to calculate errors.

Suggested improvements:

It was previously reported that CtpC is a mycobacterial P1B-type Mn²⁺ transporting ATPase (Padilla-Benavides et. al, JBC, Vol. 288, 16: 11334 –11347, 2013). However, according to a previous work performed by authors (Cell Host & Microbe 10, 248–259, September 15, 2011) CtpC was associated with resistance to toxic levels of Zn²⁺. Therefore, the role of CtpC in this work was restricted to the Zn²⁺ translocation, and transport of Mn²⁺ + across the membrane mediated by CtpC was not plenty undertaken.

In this context, it is important to know if CtpC mutants used in this work (*M. tuberculosis* and *M. smegmatis*) actually don't accumulate intracellular Mn²⁺ to definitely rule out CtpC as involved in Mn²⁺ detoxification processes. I recommend atomic absorption experiments to evaluate cation accumulation in mycobacteria.

In this work is suggested that "Pacl proteins are scaffolds that assemble P- ATPase-containing metal efflux platforms, as a novel type of functional membrane microdomain that underlies bacterial resistance to metal poisoning". Therefore, it is key to demonstrate a direct interaction between both proteins at the membrane level is important; colocalization is not enough. Coimmunoprecipitation experiments can be used in this case (Methods Mol Biol. 2017;1615:211-219. doi: 10.1007/978-1-4939-7033-9_17).

In addition, functional studies of Pacl (affinity to some type of divalent metal) are also necessary.

Minimal evidence that Pacl1 transfers metal ions to CtpC to be transported across the membrane is needed.

Even if the proposed FMMs are independent of flotillins (Curr Opin Microbiol. 2017; 36:76-84) it is necessary to have some evidence by electrophysiology about the interaction of the Pacl1-CtpC complex with typical lipids of MMFs in bacteria.

Conclusions

The performed experiments and the obtained results are consistent with the interrelation between pacl1 and CtpC, in addition to the structural details of Pacl1 to mediate the resistance to Zn²⁺ poisoning. However, direct interaction between Pacl1 and CtpC, together with functional studies of Pacl by electrophysiology are necessary to strongly suggest that the resistance to Zn²⁺ poisoning is mediated by a kind FMMs in mycobacteria.

References:

The bibliographic review is complete, robust and up-to-date

Quality of presentation:

the writing of the manuscript is adequate and the presentation of figures is excellent.

REVIEWER COMMENTS

Reviewer #1 (Remarks to the Author)

In the report, *Mycobacterial resistance to zinc poisoning requires assembly of P-ATPase-containing membrane metal efflux platforms*, Yves-Marie Boudehen et al., describes a new type of chaperone protein to cluster membrane-associated metal efflux pumps that contribute to *M. tuberculosis* infection. This referee acknowledges the amount of work and the difficulty to work with the bacterium *M. tuberculosis* in laboratory conditions. There are however several issues to address.

We thank the referee for their positive comments.

The concept of “membrane microdomain” or “membrane platform” it is not entirely clear to this referee and I think it would be important to clarify this matter. How are these concepts different from classical membrane-associated protein complexes? As far as I understand, the concept of membrane microdomain requires the coalescence of membrane lipids and proteins whereas the term “membrane platform” is rather unclear to me. It is also difficult for me to draw a line between a platform and a typical membrane-bound protein complex. The text uses indistinctly these terms and it is at times rather confusing. I did not see experiments involving membrane lipids so the authors should explain very carefully the use of the term “membrane microdomain”. Alternatively, they can perform additional experiments to show a connection between PacL1 and certain membrane lipids, to fully demonstrate that they coalesce into micro/nanodomains.

We agree with the referee that the terminology we used should be more accurate.

We initially termed the PacL1/CtpC foci “microdomains” to highlight the fact that these proteins were not evenly distributed in the plasma membrane and after a general definition given by Lucena *et al.* (BMC Biol 2018, PMID 30173665) based on their work in *B. subtilis*: “Almost 20% of membrane proteins specifically localized to the cell poles, and a vast majority of all proteins localized in distinct structures, which we term microdomains.” In *M. smegmatis*, Hayashi *et al.* (PNAS 2016, PMID 27114527) reported several membrane proteins forming foci in the plasma membrane, and also referred to these foci as “membrane domains”. As discussed by Lopez and Koch (Curr Opin Microbiol 2017 PMID 28237903) the concept of bacterial functional membrane microdomain (FMM) was originally developed after the discovery that a *B. subtilis* mutant deficient in poly-isoprenoid lipids production lost focal localization of the membrane-associated sensor kinase KinC. Bacterial microdomains are thus often compared to lipid rafts in eukaryotic cells, and the importance of specific lipids into organizing membrane microdomains in prokaryotes is discussed by Lopez and Koch. However the few examples discussed rely on indirect evidence only; in particular, whether the need for specific lipids to organize membrane proteins into bacterial FMMs is a universal rule largely remains to be explored.

To address the referee’s comment, we fractionated the total plasma membrane of recombinant *M. smegmatis* $\Delta 2$ into detergent-sensitive (DSM) and detergent-resistant (DRM) fractions and immuno-detected the Rv1488, PacL1 and CtpC proteins. This analysis showed that PacL1 and Rv1488 (FloT) partition in both the DSM and DRM fractions (see Figure below). Such partitioning pattern was previously reported for other flotillins (e.g., *B. subtilis* FloT; Scholz *et al.* 2021 Front Microbiol PMID 34777311). By contrast CtpC accumulates into the DSM fraction only. These data suggest that PacL1 might accumulate within *bona fide* “microdomains”, while the PacL1-CtpC interaction is likely detergent sensitive.

M. smegmatis $\Delta 2$ expressing PacL1-CtpC (lane 2), Rv1488::mTurquoise (lane 3), or Rv1488::mCherry-PacL1::mTurquoise-CtpC::mVenus (lane 4) were lysed and membrane fractions were separated into detergent-sensitive (DSM) and detergent-resistant (DRM) fractions using the

CellLytic™ MEM protein extraction kit (Sigma). Proteins were separated by SDS-PAGE and blotted onto nitrocellulose membrane for WB detection using an anti-GFP antibody. Lanes 1 and 5: molecular weight marker.

Future work will be needed to explore whether PacL1/CtpC foci constitute *bona fide* microdomains. To avoid confusion, we now call the observed accumulations of PacL1/CtpC in the membrane “foci” throughout our manuscript. Whether these foci represent genuine “microdomains”, *i.e.*, incorporating specific lipids, remains to be determined and has been added as a point of discussion in the revised manuscript (lines 367-75).

In our manuscript, the term “platform” referred to the function of the PacL1/CtpC protein complexes (*i.e.* in this case: metal efflux). Indeed, since we found that PacL1, CtpC, PacL2 and PacL3 cluster into the same foci, we found it appealing to call these foci “metal efflux platforms”, similar to oil rigs or oil “platforms”. Importantly, we now provide novel data showing that PacL1 physically interacts with CtpC (Fig. 5H) to form high molecular weight complexes (Fig. 5I). The text has been modified to make this point clearer (lines 85-8).

I missed infection experiments to demonstrate that mutations in PacL1 phenocopy the infection defect that the authors have reported in mutants defective in the metal efflux pumps. One would hypothesize that, as PacL1 is required for the stability of the metal pumps, a mutant in PacL1 is unable to cause an infection, as mutants in the metal pumps are infection defective mutants. I find these infection experiments important, to demonstrate there is a connection between PacL and the efflux pumps but also to demonstrate the physiological importance of PacL1 and the relevance of the formation of membrane microdomains/platforms in mycobacterial virulence.

In Botella *et al.* (Cell Host Microbe 2011, PMID 21925112), we reported that the *ctpC*-KO mutant was slightly attenuated in human macrophages (3-fold reduction in CFUs after 5 days of infection) and showed no attenuation phenotype *in vivo* in mice.

To address the referee’s comment and determine the effects of PacL1 knockout on virulence, we infected human macrophages with the *pacL1*-KO mutant and found no attenuation phenotype (data not shown). We hypothesized that this might be due to compensation by other PacL proteins. Thus, we infected human macrophages with the triple mutant KO in *pacL1*, *pacL2* and *pacL3*, a mutant that we published previously (Boudehen *et al.* 2020 Biotechniques, PMID 31937110). This mutant was not attenuated in human macrophages either (figure below).

Human monocyte-derived macrophages were infected with *M. tuberculosis* H37Rv (WT) or the triple *pacL1-2-3*-KO mutant. Cells were washed after 4 h and incubated in fresh medium at 37°C. At the indicated time-points, cells were lysed for CFU scoring on agar medium. Data show mean \pm s.d. of three biological replicates.

One explanation for this apparent discrepancy might be that *M. tuberculosis* GC1237 (lineage 2) was used in Botella *et al.*, while the less virulent H37Rv strain (lineage 4) was used in the present study. It is possible that H37Rv induces a reduced zinc burst, compared to GC1237, in macrophages, which is now briefly discussed in the revised manuscript.

Importantly, during the revision of this manuscript Mehdiratta *et al.* reported the identification of novel zinc metallophores, called “kupyaphores”, secreted by *M. tuberculosis* (Mehdiratta *et al.* 2022 PNAS, PMID 35193957). The authors show that kupyaphores are involved in both uptake of zinc in zinc-limited conditions and in resistance to zinc excess. The mechanism by which kupyaphores mediate resistance to zinc excess is not clear; yet a kupyaphore-deficient mutant is attenuated *in vivo* in mice. How kupyaphores and CtpC work together for sustaining mycobacterial resistance to zinc excess will need to be investigated; it is possible that *ctpC* deficiency in a kupyaphore-deficient background might reveal further attenuation, which is currently being explored in our laboratory through the generation of a double kupyaphore- and CtpC-deficient mutant. This is discussed in our revised manuscript (lines 317-29).

Minor comments

Are PacL2 and PacL3 structurally similar to PacL1? It seems this is not the case. Could the authors comment in the discussion section on the possible roles of these proteins?

The three proteins harbor the same DUF1490 domain, spanning almost all their amino acid sequence, except their C-terminal end. An alignment of the three proteins is shown (Fig. S1C) and highlights the conservation of the putative TM segments and the EA motifs. This is why we propose that PacL2 and PacL3 might work as metallochaperones to their cognate P-ATPases, CtpG and CtpV, respectively, like the PacL1/CtpC pair (see Discussion, lines 376-83).

The authors should include an additional control to show that the constructs to fluorescent proteins are functional and localize correctly in the bacterial membrane.

This is an excellent comment. To address it, we tested zinc sensitivity of fluorescent reporter strains and found that mTurquoise fusion to PacL1 and mVenus fusion to CtpC can confer resistance to zinc of recombinant *M. smegmatis* $\Delta 2$. These data are displayed in new Fig. S4. Regarding the localization of the fluorescent fusion proteins, our microscopy and WB data show that these fusion proteins localize to the plasma membrane as expected; whether they would co-localize with their non-tagged native forms would be very hard, if not impossible, to assess.

Fig 5C. The cells express a large amount of Flag-tagged PacL1. This may generate artefactual interactions.

This is a good point. Data shown in Fig. 5D together with new data in Fig. S4 clearly show that the level of expression of PacL1 does not impair resistance to zinc.

Reviewer #2 (Remarks to the Author)

In this manuscript, Boudehen et al. present an unusually comprehensive investigation into the function of a member of an enigmatic class of small single-pass TM proteins, denoted PacL1 (DUF1490), that is transcriptionally induced by high Zn(II) alongside the P1B-type ATPase metal effluxer, CtpC, in the human pathogen *Mycobacterium tuberculosis*. Resistance to host-mediated zinc toxicity in *M. tuberculosis* is particularly important in the phagocytic compartment in infected cells, and this is strongly worthy of detailed study. The findings reported here are novel, surprising (notably, a distant evolutionary connection to Mtb GroEL and DUF6110 proteins), and of broad significance, which should catalyze a host of follow-up studies in the bacterial metal homeostasis and resistance fields.

The work presented here sheds considerable new light on the structure and function of the PacL1 and their related orthologs in the Mtb, PacL2 and PacL3. The authors present compelling evidence in support of the hypothesis that CtpC is strictly dependent on PacL1 for functional localization to the mycobacterial membrane and metal efflux, in what could be considered dynamic microdomains. The authors tease apart and functionally segregate the domains of this ≈ 93 -residue PacL1 (with clear parallels to PacL2 and PacL3), and provide compelling evidence in support of an E/A-rich domain that is required for membrane location by PacL1, and a C-terminal Zn(II) coordination site that becomes functionally important only at high Zn(II) stress (Fig. 4D) (a striking finding). NMR studies suggest a rapidly interconverting coordination environment for the metal, which is not surprising given this short, linear metal-binding motif. They further show that PacL orthologs are at least partially redundant (but not entirely so). Finally, coincident localization of PacL1 and CtpC to the membrane occurs independent of the metal-binding site and the mycobacterial flotillin FloT, with PacL1 essentially functionally substituting for FloT.

We thank the referee for their enthusiastic comments.

I have only a few comments the authors may wish to consider:

1) p. 5, line 147: Please check the units on the radius of gyration; 3 nm is likely correct, consistent with an extended polypeptide chain.

This was a mistake indeed, which is now fixed in the revised manuscript. Thank you for having noticed it.

2) p. 7, line 232: Do *Msm* $\Delta 2$ cells expressing CtpC with PacL1 lacking E/A-rich repeats accumulate intracellular Zn as evidenced by FluoZn staining? This control appears to be missing.

Thank you for the suggestion. We performed this experiment and indeed found that complementing the *M. smegmatis* $\Delta 2$ strain with the 3EA mutant form of PacL1 results in zinc accumulation in the bacterial cytosol upon zinc stress. This is now shown in new Fig. 5F.

3) Fig. 4B, ITC: Please correct these data for the addition of Zn into buffer, and re-plot and re-fit the data. The net heat should go to 0 when the peptide is saturated with metal. Or is there another binding event here? Please indicate the fitted parameters and stand errors for this experiment (average of all experiments) in the figure legend.

We appreciate the reviewer's concern about the ITC experiments. To address the concern, we i/ calculated the average contribution of Zn/buffer interaction and show that it was of about 10% of the total signal, ii/ hypothesize about the positive contribution observed above saturating Zn concentrations, iii/ re-plotted and re-fitted the data by subtracting this response, iv/ provided the K_d and N values with error bars for the two experiments. The conclusion is that the two

experiments are i/ reproducible with K_d values around 3-4 μM and a N value around 0,7; ii/ buffer/Zn interaction only moderately (10%) contributes to the signal; iii/ it exists an endothermic contribution, not observed with peptides (Fig. S6) and that we do not attribute to a binding event.

Regarding zinc addition to the buffer, Figure 1 below displays the successive additions of 2 μL of 2 mM $\text{Zn}[\text{CH}_3\text{COO}]_2$ to 200 μL of buffer. Each addition of the metal triggers a negative response (exothermic response) of average amplitude of $-0.1 \pm 0,007 \mu\text{Watts}$ and of average area of $-0,88 \pm 0.16 \text{ kJoules}$ (as measured on the 17 last injections).

Figure 1: Zn + buffer

Regarding correction of the data for the addition of Zn into buffer: Raw data corresponding to the measurement of Zn/solPacL1 interaction performed in the same experimental conditions as those used above for (Zn+buffer) are displayed in Figure 2.

Figure 2: solPacL1 + Zn (Experiment #1). Molar ratios are indicated above or below the pics.

A correction for the “Zn+buffer” signal would only moderately modify the raw data. If we consider its average amplitude, it would correspond to the subtraction of $-0.1 \mu\text{Watts}$ to negative pics reaching up $-1 \mu\text{Watts}$ and to positive pics reaching about $0.7 \mu\text{Watts}$.

Obviously, applying such a correction will not raise the raw signal to zero at zinc concentrations above the saturating concentration (i.e. above a molar ratio of 1). At higher Zn concentrations a positive (endothermic) response is still present even taking in consideration the contribution of “Zn+ buffer”.

As shown in Figure 3 below, the same observation can be made in the second experiment. Note that this experiment has been performed in the same experimental conditions as those used previously except that injections of 2 μL contained 1.5 instead of 2 mM $\text{Zn}[\text{CH}_3\text{COO}]_2$.

Figure 3: solPacL1 + Zn (Experiment #2). Molar ratios are indicated above or below the pics.

So far, we do not have any explanation for this phenomenon but we observed that it is not present in the experiments with peptides (Fig S6D).

Calculation of curve: kJ mol^{-1} of injectant = $f(\text{molar ratio})$

If we assume that the positive response observed at high Zn concentrations does not correspond to Zn binding to solPacL1, and is present all along the experiment but masked by the exergonic reaction of zinc binding to solPacL1, we can subtract its average area (Kjoule) to build the curve kJ mol^{-1} of injectant = $f(\text{molar ratio})$ (Figure 4 below).

Figure 4. Upper panel: experiment #1; lower panel: experiment #2.

Data were fitted with a single binding model. Data of experiment #1 are fitted with a K_d of $3.2 \pm 1 \mu\text{M}$ and N (sites) of 0.719 ± 0.018 . Data of experiment #2 are fitted with a K_d of $3.6 \pm 0.5 \mu\text{M}$ and N (sites) of 0.691 ± 0.009 .

Figure 5 below displays the two experiments on the same graph.

Figure 5

New panels are provided in Fig. 4B and S6D, which addresses the referee's concern.

4) p. 6, line 217: Please provide a reference after the word "metallochaperone"

Thank you for catching this - Fu *et al.* 2013 Nat Chem Biol is now cited.

5) Fig. 6D, line 1045: I find it puzzling that the authors did not do a parallel series of experiments with the PacL3. Why not? One might anticipate full Cu-dependent complementation and rescue from Zn toxicity in the $\Delta pacL1$ strain.

Indeed this is an exciting hypothesis that we now assess experimentally. However, neither 5 nor 50 μM CuSO_4 restored growth of the $\Delta pacL1$ *M. tuberculosis* strain in the presence of 100 μM ZnSO_4 . There could be many explanations for this. Unlike cadmium, copper can exist in two oxidation states, which can vary depending on the experimental conditions. Alternatively, PacL3 may be insufficiently induced by copper under these experimental conditions for other reasons, due to regulatory differences among the *pacL* promoters or Zn-Cu crosstalk. All of these possibilities would require significant further experimental investigation, which we believe would be a departure from the main narrative of our paper, and we thus prefer to leave these questions to future work. This is now briefly discussed in the revised manuscript (lines 297-301).

6) What's left for future work is a mechanistic unpacking what appears to be co-localization phenomenon of the PacL1 and CtpC. In my mind the authors provide no strong support for or against a metallochaperone model (direct metal transfer by ligand exchange), and the absence of detection of even a transient physical interaction between the two proteins, would seem to argue for a local metal concentration or sequestration effect. Perhaps the discussion section can be revised to better reflect these ideas.

We thank the reviewer for this comment, which was also echoed by other reviewers. To address this, we added new split-GFP experiments using a bipartite strategy (instead of a tripartite version in the previous manuscript) and different linkers, which allowed us to detect an interaction between PacL1 and the N-terminal region of CtpC, which contains its putative metal-binding domain (Figures 5H and S9). Furthermore, we conducted Blue-native PAGE analysis, which indicated that CtpC and PacL1 interact at the plasma membrane, forming high-molecular weight complexes (Figure 5I). Whether and how PacL1-CtpC interaction sustains CtpC stabilization and/or metal transfer from PacL1 to CtpC will need further investigation. This is now briefly discussed in our revised manuscript (lines 367-75).

7) Can the authors comment on the mode of Zn-inducible transcription of the *pacL1-ctpC*?

Unlike the *pacL2/ctpG* and *pacL3/ctpV* operons, the *pacL1/ctpC* operon is not located downstream from a transcriptional regulator-encoding gene. Thus, it is difficult to elaborate on a possible regulation scenario. We can hypothesize that SmtB/Rv2358 might be involved based on findings in *M. smegmatis* (Goethe *et al.* 2020 mSystems PMID 32317393). However, zinc resistance in *M. smegmatis* is mostly sustained by (SmtB-regulated) ZitA, which is not the case in *M. tuberculosis*. Whether PacL1/CtpC regulation in *M. tuberculosis* is mediated by SmtB or by one or more other zinc-dependent transcriptional regulators remains to be elucidated. We would prefer avoiding discussing this aspect in the present manuscript, which would rely on far too speculative considerations and will surely be the subject of future work.

Reviewer #3 (Remarks to the Author)

Manuscript: NCOMMS-21-35884-T
Reviewer comments

Key results:

This work addresses a relevant topic in mycobacteria: the molecular mechanisms activated by *M. tuberculosis* to overcome toxic levels of heavy metals inside macrophages. The possible role of some P-type ATPases in heavy metal detoxification processes and ion specificity to exert mycobacterial resistance to metal poisoning is known. However, there is lack of information about the correlation of P-type ATPases with others cell components, such as metallochaperones, involved in the ion homeostasis of the plasma membrane.

The results presented in this work are valuable because, for the first time, other membrane components are associated with the P-type ATPase activity (in this case mediated by CtpC) in the mycobacterial cell membrane is evidenced. The existence of this putative new family of proteins (PacLs) is postulated, being part of the same operon of different mycobacterial P-type ATPases: CtpC, CtpG or CtpG.

Specifically, this work suggests that the PacL1 interaction with CtpC is necessary to prevent Zn²⁺ accumulation inside mycobacteria. In addition, structural elements in PacL1 necessary for the CtpC activation and function to exert mycobacterial resistance to Zn²⁺ poisoning are even described. The presence of flotillin-independent FMMs that underlies mycobacterial resistance to metal poisoning is proposed, a novel mechanism at date unknown for mycobacteria.

Validity:

This manuscript has a correct execution; therefore, I consider there is no reason to prohibit its publication.

Originality:

According to the literature I have checked to date, the conception of this work is original, together with its execution, results, discussion and conclusions.

We would like to thank the referee for these positive comments.

Significance:

Since much information is still lacking of the mycobacterial P-type ATPases functioning, I sincerely consider that the results presented in this work are interesting and valuable. Since MMFs have already been described in bacteria, there is no urgency in the publication of this manuscript. In addition, it seems that the mycobacterial resistance to Zn²⁺ poisoning and this cation translocation across the cell membrane is not exclusive to P-type ATPases.

The referee is right. As outlined in our manuscript resistance to zinc excess in *M. smegmatis* relies on the permease ZitA, whereas resistance to zinc excess in *M. tuberculosis* relies mostly on the P-type ATPase CtpC.

Abstract:

The text of the abstract is clear, however. The main purpose of the manuscript is to describe and report a new type of protein, which is not fully included in the abstract.

We took this comment into account in our revised manuscript (see abstract).

Introduction:

Although the main objective of the study is apparently to describe the PacLs proteins in mycobacteria, the same study links those closely to the function of the mycobacterial cell membrane transporter CtpC. In this sense, it is strange that there is no reference to this topic in the introduction (Padilla-Benavides *et al.*, JBC, Vol. 288, 16: 11334 –11347, 2013) where in a elegant way, it is demonstrated that Mn²⁺ or Ni²⁺ (to a lesser extent) can be CtpC substrates in mycobacterial cells. It is necessary to clarify which of these two cations is the main CtpC substrate *in vitro* and *in vivo*, in *M. tuberculosis* and *M. smegmatis*, according to the known information; additionally, to clarify the possible role of CtpC in the mycobacterial resistance to toxic Zn²⁺ concentrations (Cell Host & Microbe 10, 248–259, September 15, 2011). To begin with, it is very relevant evidence that must be taken as part of the approach of this work.

We apologize for any confusion, however the reference mentioned by the referee (Padilla-Benavides *et al.* 2013 J Biol Chem) is already cited in our introduction (lines 65-9) and the findings of that study are discussed in our Discussion section (lines 330-41).

In brief, Padilla-Benavides *et al.* found that CtpC transports manganese, although at an unusually slow rate. Of note, PacL1 was not taken into account in that study. Our new data clearly show that PacL1/CtpC does not sustain mycobacterial resistance to manganese excess, and provide sufficient evidence that zinc is the main substrate functionally in cells, which is more important than *in vitro* work. As mentioned in the manuscript, we were not able to purify CtpC in sufficient quantities and quality to address this biochemically in the current study. Yet, we have transparently discussed this *priori* work and addressed the Mn vs. Zn issue experimentally.

Data & methodology:

I consider the data reported and the methodology presented in this manuscript are sufficiently detailed and clear to allow the reproduction of the results.

The strategy proposed to solve the different research questions is adequate; the experiments are sound, and usually the data obtained are reliable. Here a description of the most relevant experiments according to the reviewer's point of view.

The western blot and kinetic of growth experiments showed in figure 2 clearly demonstrate that the PacL1 and CtpC expression in *M. tuberculosis* and *M. smegmatis* is mediated by Zn²⁺. In addition, both, PacL1 and CtpC are necessary for Zn²⁺ efflux under toxic concentrations of this cation, being essential the TM domain of PacL1. Western blotting, Flow cytometry and Eclipse TI-E/B wide field epifluorescence experiments in Figure 2 and 3, also demonstrate the colocalization of PacL1 and CtpC in the plasma membranes of *M. tuberculosis* and *M. smegmatis*.

Regarding to the binding stoichiometry of PacL1/Zn²⁺, the equilibrium dialysis and ITC experiments partially demonstrate that PacL1 binds Zn²⁺ (not Mn²⁺) in a 1:1 molar ratio, and mediated by Hys residues at the MBM of PacL1. This finding is also corroborated by NMR spectroscopy. Therefore, it is clear that PacL1 MBM is required for mycobacterial resistance to zinc at higher concentrations.

Preliminary evidence that PacL1 could be a flotillin-like protein is observed in immunofluorescence, immunoblotting and qRT-PCR experiments showed in Figures S5 and S7. Specifically, results suggest that the Glu/Ala repeats in PacL1 cytoplasmic region is required for abundant CtpC levels in the plasma membrane, and for the proper function of PacL1 triggering resistance to Zn²⁺ excess. The immunofluorescence experiments also demonstrate that PacL2 and PacL3 colocalize in the cell membrane and are associated to bacterial resistance to metal poisoning.

Detailing the PacL1 portion involved in the Zn²⁺ resistance, the immunofluorescent label in Figure 6 indicates that the PacL1 MBM is required for resistance to high concentration of Zn²⁺. In addition, that CtpC is necessary (I would not say essential) for Zn²⁺ resistance.

We thank the referee for their positive comments.

In response to their last point, the term “Essential” has been exchanged for “necessary”, line 312.

Use of statistics and treatment of uncertainties:

Although the number of processed samples in some experiments allows a clear statistical significance (e.g. Fig 1E, 6E), by contrast, the error from bars is not quantified in other cases (e.g. Fig 1B, 1G, 3I, 4D, 5A, 5D, 5E, 6D, etc). On the other hand, the respective standard deviation for values obtained by means, always should be reported. For this reason, the number of dates obtained from processed samples have to be increased to calculate errors.

We thank the referee for this comment, and we have improved the statistical analyses and their presentation throughout the manuscript as follows:

In Fig. 1B, since the data are “all-or-nothing”, we are not convinced they do need statistical analysis. If the referee still thinks that some sort of analysis is needed here, we would like to ask him/her for clarification and we would be happy to add this analysis before publication.

Statistical analysis is provided in Fig. 1G.

In Fig. 3I, again there is no difference between zinc sensitivity of the WT and mutant strains at 250 μ M zinc; in other words the red curves are superimposable, which we now emphasize by including the dotted line in the right panel.

In Fig. 4D, the error bars are present; however they are small and might be hard to visualize (see magnification of the upper panel below). Again, since the data are “all-or-nothing”, we are not convinced they do need statistical analysis. If the referee still thinks that some sort of analysis is needed here, we would like to ask him/her for clarification and we would be happy to add this analysis before publication.

In Fig. 5A, the data show quantification of one WB (displayed on the left), and are representative of two independent experiments. This is now clarified in the figure legend.

In Fig. 5D and 5E, since the data are “all-or-nothing”, we are not convinced they do need statistical analysis. If the referee still thinks that some sort of analysis is needed here, we would like to ask him/her for clarification and we would be happy to add this analysis before publication.

In Fig. 6D, indeed, error bars were not visible in the right panel; the scale has been modified so that they are now visible. Data in the middle panel are now analyzed statistically using Mann-Whitney.

Suggested improvements:

It was previously reported that CtpC is a mycobacterial P1B-type Mn²⁺ transporting ATPase (Padilla-Benavides et. al, JBC, Vol. 288, 16: 11334 –11347, 2013). However, according to a previous work performed by authors (Cell Host & Microbe 10, 248–259, September 15, 2011) CtpC was associated with resistance to toxic levels of Zn²⁺. Therefore, the role of CtpC in this work was restricted to the Zn²⁺ translocation, and transport of Mn²⁺ + across the membrane mediated by CtpC was not plenty undertaken.

In this context, it is important to know if CtpC mutants used in this work (*M. tuberculosis* and *M. smegmatis*) actually don't accumulate intracellular Mn²⁺ to definitely rule out CtpC as involved in Mn²⁺ detoxification processes. I recommend atomic absorption experiments to evaluate cation accumulation in mycobacteria.

To address this comment, we evaluated the sensitivity of the $\Delta 2$ mutant strain, with or without expression of PacL1 and CtpC, to manganese in liquid medium. Our data (now displayed in Fig. S2H) convincingly demonstrate that PacL1/CtpC do not sustain mycobacterial resistance to manganese excess. We believe that these *in vivo* results are the most physiologically relevant way to demonstrate the role of PacL1 and CtpC in *M. tuberculosis*, and therefore we did not view it as necessary to spend a great deal of effort to optimize purification of CtpC to attempt to confirm these findings *in vitro*. This work is currently ongoing in our laboratory, and we feel it is beyond the scope of the present study.

In this work is suggested that “PacL proteins are scaffolds that assemble P- ATPase-containing metal efflux platforms, as a novel type of functional membrane microdomain that underlies bacterial resistance to metal poisoning”. Therefore, it is key to demonstrate a direct interaction between both proteins at the membrane level is important; colocalization is not enough. Coimmunoprecipitation experiments can be used in this case (Methods Mol Biol. 2017;1615:211-219. doi: 10.1007/978-1-4939-7033-9_17).

Thank you for pointing this out – we completely agree, and this issue was raised by another referee as well. To address this concern, additional split-GFP experiments using a bipartite strategy (instead of a tripartite version in the original manuscript) and different linkers allowed us to detect an interaction between PacL1 and the N-terminal region of CtpC, which contains its putative metal-binding domain (Figures 5H and S9). Furthermore, Blue-native PAGE indicated that CtpC and PacL1 interact at the plasma membrane, forming high-molecular weight complexes (Figure 5I). Whether and how PacL1-CtpC interaction sustains CtpC stabilization and/or metal transfer from PacL1 to CtpC will need further investigation. This is now discussed in our revised manuscript (lines 367-75).

In addition, functional studies of PacL (affinity to some type of divalent metal) are also necessary.

This is a great point. We have performed experiments indicating that PacL1 can bind Ni²⁺ with a similar affinity as for Zn²⁺, as well as bind Cd²⁺ with very low affinity (100 μ M). On the other hand, PacL1 does not detectably bind Fe²⁺ (data not shown). These data are part of an ongoing follow up study in our lab and, while you would prefer to hold the data for future publication as part of that story, we defer to the referee and editor if they think it is necessary to include in the current manuscript.

Minimal evidence that PacL1 transfers metal ions to CtpC to be transported across the membrane is needed.

We agree that exploring this aspect will be very interesting; however, without currently being able to purify both PacL1 and CtpC at sufficient levels to conduct *in vitro* biochemical and biophysical work, it will require significant further work and optimization in order to conduct mechanistic analysis at this molecular level. We believe that, already, our manuscript provides a broad scope of different analyses from cellular to structural, which is in agreement with the positive comments

from all referees about the comprehensive nature of our study. While we agree that further mechanistic questions remain, it is our view that this falls beyond the scope of the current manuscript. In fact, we believe that a major strength of this work is to prompt future research in this area by many groups.

Even if the proposed FMMs are independent of flotillins (Curr Opin Microbiol. 2017; 36:76-84) it is necessary to have some evidence by electrophysiology about the interaction of the PacL1-CtpC complex with typical lipids of MMFs in bacteria.

We recognize that our language characterizing the PacL1/CtpC platforms as “microdomains” may have been overzealous in the absence of strong evidence to show association of particular lipids with these foci. Please see our response to referee #1’s first major comment for more details – we have opted to change our terminology to describe these assemblies as “foci” rather than “microdomains”.

Considering the referee’s suggestion to use electrophysiology to probe this complex:

Since we could not purify CtpC or PacL1/CtpC, we could not measure ATPase activity nor phosphorylation activity, which are the easiest techniques to implement on an ATPase. As a consequence, nothing is known about the parameters that govern the functioning of CtpC, nor its mechanism, including electrogenicity for example. In addition, electrophysiology was initially developed for ion channels that transport 10^7 to 10^8 ions/sec. However, even a very active ATPase such as the Na^+/K^+ -ATPase transports “only” 10^2 to 10^3 ions/sec. Thus, very large amounts of (possibly purified) ATPase and a “watertight” system would be needed to correctly measure such low fluxes. We are a long way from that with CtpC. Finally, to our knowledge, electrophysiology on bacterial membranes is much more difficult to perform than on eukaryotic membranes due to the presence of the cell wall. This why the most used model for electrophysiology is the xenopus oocyte.

Conclusions

The performed experiments and the obtained results are consistent with the interrelation between pacL1 and CtpC, in addition to the structural details of PacL1 to mediate the resistance to Zn^{2+} poisoning. However, direct interaction between PacL1 and CtpC, together with functional studies of PacL by electrophysiology are necessary to strongly suggest that the resistance to Zn^{2+} poisoning is mediated by a kind FMMs in mycobacteria.

Thank you for these comments, which we believe we have fully addressed in our responses to the referees’ comments above. In summary, we now provide evidence for specific (3EA-mediated) interactions between PacL1 and CtpC, and unfortunately the proposed electrophysiology studies are not experimentally feasible in this system. Extensive biophysical studies of the three PacL/Ctp systems is currently ongoing in our laboratory, representing years’ worth of work that goes far beyond the scope of this manuscript and its conclusions.

References:

The bibliographic review is complete, robust and up-to-date.

Quality of presentation:

the writing of the manuscript is adequate and the presentation of figures is excellent.

We thank the referee for this positive appraisal.

Reviewers' Comments:

Reviewer #1:

Remarks to the Author:

The authors addressed my comments adequately.

It is interesting that Rv1488 partitioned into the DRM fraction, pointing to the existence of a liquid-ordered phase in the membranes and the colocalization with a flotillin-like protein. It is intriguing that the flotillin interacting protein partner does not localize in these regions. Maybe these interactions are transient.

It is also intriguing that the PacL1 mutant shows no attenuation in virulence, as is now discussed in this manuscript.

Reviewer #2:

Remarks to the Author:

The authors have adequately addressed all of the comments that I raised in the initial review to my satisfaction. The addition of the new data in Figs. 4 and 5 provide additional support for the author's model. This manuscript represents a highly significant advance in our field. The requested infection data, while negative, do little to diminish my enthusiasm for what will be a very important paper on a very important microbial pathogen.

Reviewer #3:

Remarks to the Author:

Review of revised Nature Communications manuscript NCOMMS-21-35884A

Below, my point of view of the answers given by the author about my concerns.

My concern:

The text of the abstract is clear, however. The main purpose of the manuscript is to describe and report a new type of protein, which is not fully included in the abstract.

author's response:

We took this comment into account in our revised manuscript (see abstract).

My point of view about the answer:

the author's answer is satisfactory

My concern:

Although the main objective of the study is apparently to describe the PaLs proteins in mycobacteria, the same study links those closely to the function of the mycobacterial cell membrane transporter CtpC. In this sense, it is strange that there is no reference to this topic in the introduction (Padilla-Benavides et. al, JBC, Vol. 288, 16: 11334 –11347, 2013) where in a elegant way, it is demonstrated that Mn²⁺ or Ni²⁺ (to a lesser extent) can be CtpC substrates in mycobacterial cells. It is necessary to clarify which of these two cations is the main CtpC substrate in vitro and in vivo, in *M. tuberculosis* and *M. smegmatis*, according to the known information; additionally, to clarify the possible role of CtpC in the mycobacterial resistance to toxic Zn²⁺ concentrations (Cell Host & Microbe 10, 248–259, September 15, 2011). To begin with, it is very relevant evidence that must be taken as part of the approach of this work.

author's response:

We apologize for any confusion, however the reference mentioned by the referee (Padilla-Benavides et al. 2013 J Biol Chem) is already cited in our introduction (lines 65-9) and the findings

of that study are discussed in our Discussion section (lines 330-41).

In brief, Padilla-Benavides et al. found that CtpC transports manganese, although at an unusually slow rate. Of note, PacL1 was not taken into account in that study. Our new data clearly show that PacL1/CtpC does not sustain mycobacterial resistance to manganese excess, and provide sufficient evidence that zinc is the main substrate functionally in cells, which is more important than in vitro work. As mentioned in the manuscript, we were not able to purify CtpC in sufficient quantities and quality to address this biochemically in the current study. Yet, we have transparently discussed this priori work and addressed the Mn vs. Zn issue experimentally.

My point of view about the answer:
the author's answer is satisfactory

My concern:

Detailing the PacL1 portion involved in the Zn²⁺ resistance, the immunofluorescent label in Figure 6 indicates that the PacL1 MBM is required for resistance to high concentration of Zn²⁺. In addition, that CtpC is necessary (I would not say essential) for Zn²⁺ resistance.

author's response:

We thank the referee for their positive comments.

In response to their last point, the term "Essential" has been exchanged for "necessary", line 312.

My point of view about the answer:
the author's answer is satisfactory

My concern:

Use of statistics and treatment of uncertainties:

Although the number of processed samples in some experiments allows a clear statistical significance (e.g. Fig 1E, 6E), by contrast, the error from bars is not quantified in other cases (e.g. Fig 1B, 1G, 3I, 4D, 5A, 5D, 5E, 6D, etc). On the other hand, the respective standard deviation for values obtained by means, always should be reported. For this reason, the number of dates obtained from processed samples have to be increased to calculate errors.

author's response:

We thank the referee for this comment, and we have improved the statistical analyses and their presentation throughout the manuscript as follows:

In Fig. 1B, since the data are "all-or-nothing", we are not convinced they do need statistical analysis. If the referee still thinks that some sort of analysis is needed here, we would like to ask him/her for clarification and we would be happy to add this analysis before publication.

Statistical analysis is provided in Fig. 1G.

In Fig. 3I, again there is no difference between zinc sensitivity of the WT and mutant strains at 250 μ M zinc; in other words the red curves are superimposable, which we now emphasize by including the dotted line in the right panel.

In Fig. 4D, the error bars are present; however they are small and might be hard to visualize (see magnification of the upper panel below). Again, since the data are "all-or-nothing", we are not convinced they do need statistical analysis. If the referee still thinks that some sort of analysis is needed here, we would like to ask him/her for clarification and we would be happy to add this analysis before publication.

In Fig. 5A, the data show quantification of one WB (displayed on the left), and are representative of two independent experiments. This is now clarified in the figure legend.

In Fig. 5D and 5E, since the data are "all-or-nothing", we are not convinced they do need statistical analysis. If the referee still thinks that some sort of analysis is needed here, we would like to ask him/her for clarification and we would be happy to add this analysis before publication.

In Fig. 6D, indeed, error bars were not visible in the right panel; the scale has been modified so that they are now visible. Data in the middle panel are now analyzed statistically using Mann-

Whitney.

My point of view about the answer:

The reviewer considers that the author's response is adequate, and the statistical treatment of the results is well explained now.

My concern:

It was previously reported that CtpC is a mycobacterial P1B-type Mn²⁺ transporting ATPase (Padilla-Benavides et. al, JBC, Vol. 288, 16: 11334 –11347, 2013). However, according to a previous work performed by authors (Cell Host & Microbe 10, 248–259, September 15, 2011) CtpC was associated with resistance to toxic levels of Zn²⁺. Therefore, the role of CtpC in this work was restricted to the Zn²⁺ translocation, and transport of Mn²⁺ across the membrane mediated by CtpC was not plenty undertaken.

In this context, it is important to know if CtpC mutants used in this work (*M. tuberculosis* and *M. smegmatis*) actually don't accumulate intracellular Mn²⁺ to definitely rule out CtpC as involved in Mn²⁺ detoxification processes. I recommend atomic absorption experiments to evaluate cation accumulation in mycobacteria.

author's response:

To address this comment, we evaluated the sensitivity of the $\Delta 2$ mutant strain, with or without expression of Pacl1 and CtpC, to manganese in liquid medium. Our data (now displayed in Fig. S2H) convincingly demonstrate that Pacl1/CtpC do not sustain mycobacterial resistance to manganese excess. We believe that these in vivo results are the most physiologically relevant way to demonstrate the role of Pacl1 and CtpC in *M. tuberculosis*, and therefore we did not view it as necessary to spend a great deal of effort to optimize purification of CtpC to attempt to confirm these findings in vitro. This work is currently ongoing in our laboratory, and we feel it is beyond the scope of the present study.

My point of view about the answer:

According to the stated in the new version of the manuscript, I'm partially agree with the author response. I accept that functional studies have greater validity using in vivo models than in vitro ones. In this manuscript (figures 1E, 1F, 1G) complementation of the *M. smegmatis* $\Delta 2$ mutant with Pacl1 and CtpC is evidenced. However, the same does not happen with *M. tuberculosis* mutants. In the results (line 123-135) the authors stated that "We generated recombinant $\Delta 2$ mutants expressing untagged or epitope- Tagged versions of *M. tuberculosis* Pacl1 and CtpC under the control of their native, zinc-inducible, promoter (Ppacl1) (Figure S2E).) as in *M. tuberculosis* (12), and both Pacl1 and CtpC proteins were correctly expressed in the recombinant strains (Figure S2G)". However, in the included reference (Cell Host Microbe 10, 248-259 (2011), Mn²⁺ is not used to stress *M. tuberculosis* mutants defective in CtpC. A possible solution would be to use mutants of *M. tuberculosis* H37Rv complemented with CtpC to evaluate mycobacteria susceptibility to toxic concentrations of Manganese.

My concern:

In this work is suggested that "Pacl proteins are scaffolds that assemble P- ATPase-containing metal efflux platforms, as a novel type of functional membrane microdomain that underlies bacterial resistance to metal poisoning". Therefore, it is key to demonstrate a direct interaction between both proteins at the membrane level is important; colocalization is not enough. Coimmunoprecipitation experiments can be used in this case (Methods Mol Biol. 2017;1615:211-219. doi: 10.1007/978-1-4939-7033-9_17).

author's response:

Thank you for pointing this out – we completely agree, and this issue was raised by another referee as well. To address this concern, additional split-GFP experiments using a bipartite strategy (instead of a tripartite version in the original manuscript) and different linkers allowed us to detect an interaction between Pacl1 and the N-terminal region of CtpC, which contains its putative metal-binding domain (Figures 5H and S9). Furthermore, Blue-native PAGE indicated that CtpC and

PacL1 interact at the plasma membrane, forming high-molecular weight complexes (Figure 5I). Whether and how PacL1-CtpC interaction sustains CtpC stabilization and/or metal transfer from PacL1 to CtpC will need further investigation. This is now discussed in our revised manuscript (lines 367-75).

My point of view about the answer:

The reviewer agrees with the author's response

My concern:

Minimal evidence that PacL1 transfers metal ions to CtpC to be transported across the membrane is needed.

author's response:

We agree that exploring this aspect will be very interesting; however, without currently being able to purify both PacL1 and CtpC at sufficient levels to conduct in vitro biochemical and biophysical work, it will require significant further work and optimization in order to conduct mechanistic analysis at this molecular level. We believe that, already, our manuscript provides a broad scope of different analyses from cellular to structural, which is in agreement with the positive comments from all referees about the comprehensive nature of our study. While we agree that further mechanistic questions remain, it is our view that this falls beyond the scope of the current manuscript. In fact, we believe that a major strength of this work is to prompt future research in this area by many groups.

My point of view about the answer:

Since the author considers that it is laborious to obtain active proteins of this type to carry out functional studies, I agree with the author's answer, therefore, these data could be included in a new study focused on molecular mechanisms of ion transfer from PacL1 to CtpC.

My concern:

In addition, functional studies of PacL (affinity to some type of divalent metal) are also necessary.

author's response:

This is a great point. We have performed experiments indicating that PacL1 can bind Ni²⁺ with a similar affinity as for Zn²⁺, as well as bind Cd²⁺ with very low affinity (100 μ M). On the other hand, PacL1 does not detectably bind Fe²⁺ (data not shown). These data are part of an ongoing follow up study in our lab and, while you would prefer to hold the data for future publication as part of that story, we defer to the referee and editor if they think it is necessary to include in the current manuscript.

My point of view about the answer:

The reviewer considers that part of the results that the author mentioned above should be included in this manuscript to evidence that PacL1 binds divalent cations.

My concern:

Even if the proposed FMMs are independent of flotillins (Curr Opin Microbiol. 2017; 36:76-84) it is necessary to have some evidence by electrophysiology about the interaction of the PacL1-CtpC complex with typical lipids of MMFs in bacteria.

author's response:

We recognize that our language characterizing the PacL1/CtpC platforms as "microdomains" may have been overzealous in the absence of strong evidence to show association of particular lipids with these foci. Please see our response to referee #1's first major comment for more details – we have opted to change our terminology to describe these assemblies as "foci" rather than "microdomains".

Considering the referee's suggestion to use electrophysiology to probe this complex:

Since we could not purify CtpC or PacL1/CtpC, we could not measure ATPase activity nor

phosphorylation activity, which are the easiest techniques to implement on an ATPase. As a consequence, nothing is known about the parameters that govern the functioning of CtpC, nor its mechanism, including electrogenicity for example. In addition, electrophysiology was initially developed for ion channels that transport 10⁷ to 10⁸ ions/sec. However, even a very active ATPase such as the Na⁺/K⁺-ATPase transports "only" 10² to 10³ ions/sec. Thus, very large amounts of (possibly purified) ATPase and a "watertight" system would be needed to correctly measure such low fluxes. We are a long way from that with CtpC. Finally, to our knowledge, electrophysiology on bacterial membranes is much more difficult to perform than on eukaryotic membranes due to the presence of the cell wall. This why the most used model for electrophysiology is the xenopus oocyte.

My point of view about the answer:

The reviewer agrees with the author's response

My concern:

The performed experiments and the obtained results are consistent with the interrelation between pacl1 and CtpC, in addition to the structural details of Pacl1 to mediate the resistance to Zn²⁺ poisoning. However, direct interaction between Pacl1 and CtpC, together with functional studies of Pacl by electrophysiology are necessary to strongly suggest that the resistance to Zn²⁺ poisoning is mediated by a kind FMMs in mycobacteria.

author's response:

Thank you for these comments, which we believe we have fully addressed in our responses to the referees' comments above. In summary, we now provide evidence for specific (3EA-mediated) interactions between Pacl1 and CtpC, and unfortunately the proposed electrophysiology studies are not experimentally feasible in this system. Extensive biophysical studies of the three Pacl/Ctp systems is currently ongoing in our laboratory, representing years' worth of work that goes far beyond the scope of this manuscript and its conclusions.

My point of view about the answer:

The reviewer considers that now there is evidence for direct interaction between Pacl and CtpC, which can be accepted in the new version of the manuscript.

REVIEWER COMMENTS

We thank referees #1 and #2 for their positive comments.

Reviewer #3 (Remarks to the Author)

Review of revised Nature Communications manuscript NCOMMS-21-35884A

Below, my point of view of the answers given by the author about my concerns.

We thank the referee for their positive comments.

My concern:

It was previously reported that CtpC is a mycobacterial P1B-type Mn²⁺ transporting ATPase (Padilla-Benavides et. al, JBC, Vol. 288, 16: 11334 –11347, 2013). However, according to a previous work performed by authors (Cell Host & Microbe 10, 248–259, September 15, 2011) CtpC was associated with resistance to toxic levels of Zn²⁺. Therefore, the role of CtpC in this work was restricted to the Zn²⁺ translocation, and transport of Mn²⁺ across the membrane mediated by CtpC was not plenty undertaken.

In this context, it is important to know if CtpC mutants used in this work (*M. tuberculosis* and *M. smegmatis*) actually don't accumulate intracellular Mn²⁺ to definitely rule out CtpC as involved in Mn²⁺ detoxification processes. I recommend atomic absorption experiments to evaluate cation accumulation in mycobacteria.

author's response:

To address this comment, we evaluated the sensitivity of the $\Delta 2$ mutant strain, with or without expression of PacL1 and CtpC, to manganese in liquid medium. Our data (now displayed in Fig. S2H) convincingly demonstrate that PacL1/CtpC do not sustain mycobacterial resistance to manganese excess. We believe that these in vivo results are the most physiologically relevant way to demonstrate the role of PacL1 and CtpC in *M. tuberculosis*, and therefore we did not view it as necessary to spend a great deal of effort to optimize purification of CtpC to attempt to confirm these findings in vitro. This work is currently ongoing in our laboratory, and we feel it is beyond the scope of the present study.

My point of view about the answer:

According to the stated in the new version of the manuscript, I'm partially agree with the author response. I accept that functional studies have greater validity using in vivo models that in vitro ones. In this manuscript (figures 1E, 1F, 1E) complementation of the *M. smegmatis* $\Delta 2$ mutant with PacL1 and CtpC is evidenced. However, the same does not happen with *M. tuberculosis* mutants. In the results (line 123-135) the authors stated that "We generated recombinant $\Delta 2$ mutants expressing untagged or epitope-tagged versions of *M. tuberculosis* PacL1 and CtpC under the control of their native, zinc-inducible, promoter (PpacL1) (Figure S2E).) as in *M. tuberculosis* (12), and both PacL1 and CtpC proteins were correctly expressed in the recombinant strains (Figure S2G)". However, in the included reference (Cell Host Microbe 10, 248-259 (2011), Mn²⁺ is not used to stress *M. tuberculosis* mutants defective in CtpC. A possible solution would be to use mutants of *M. tuberculosis* H37Rv complemented with CtpC to evaluate mycobacteria susceptibility to toxic concentrations of Manganese.

We appreciate the referee's comment, and will certainly take it into account in our future functional studies aiming at deciphering the mechanism of PacL-Ctp-mediated metal transport.

My concern:

In addition, functional studies of PacL (affinity to some type of divalent metal) are also necessary.

author's response:

This is a great point. We have performed experiments indicating that PacL1 can bind Ni²⁺ with a similar affinity as for Zn²⁺, as well as bind Cd²⁺ with very low affinity (100 μ M). On the other hand, PacL1 does not detectably bind Fe²⁺ (data not shown). These data are part

of an ongoing follow up study in our lab and, while you would prefer to hold the data for future publication as part of that story, we defer to the referee and editor if they think it is necessary to include in the current manuscript.

My point of view about the answer:

The reviewer considers that part of the results that the author mentioned above should be included in this manuscript to evidence that PacL1 binds divalent cations.

We appreciate the referee's comment but we would rather prefer consolidating these data for a future follow-up manuscript.